# TRAJECTORY OPTIMAL ANISOTROPIC DIFFUSION MODELS

## ABSTRACT

We study anisotropic diffusion for generative modeling by replacing the scalar noise schedule with a matrix-valued path $M_t$ that allocates noise (and denoising effort) across subspaces. We introduce a trajectory-level objective that jointly trains the score network and *learns* $M_t(\theta)$; in the isotropic case, it recovers standard score matching, making schedule learning equivalent to choosing the weight over noise levels. We further derive an efficient estimator for $\partial_\theta \nabla \log p_t$ that enables efficient optimization of $M_t$. For inference, we develop an anisotropic reverse-ODE sampler based on a second-order Heun update with a closed-form step, and we learn a scalar time-transform $r(t; \gamma)$ that targets discretization error. Across CIFAR-10, AFHQv2, and FFHQ, our method matches EDM overall and substantially improve few-step generation. Together, these pieces yield a practical, trajectory-optimal recipe for anisotropic diffusion. Code is available at [1].

## 1 INTRODUCTION

Diffusion and flow-based generative models typically add and remove *isotropic* Gaussian noise with a scalar schedule $\sigma(t)$ while learning a score network and integrating a reverse-time SDE/ODE (Ho et al., 2020; Song et al., 2021; Karras et al., 2022). The isotropic design is simple and effective, but forces the dynamics to act uniformly in all directions.

**Why anisotropy.** Replacing the scalar schedule by a matrix-valued path $M_t$ substantially enlarges the design space: noise can be allocated differently across subspaces and time, better matching data geometry–natural images concentrate energy in low spatial frequencies (Ruderman & Bialek, 1993); latent diffusion offloads fine detail to a learned autoencoder (Rombach et al., 2022); video models benefit from temporally structured priors or decomposed noise (Ge et al., 2023; Luo et al., 2023); multi-resolution autoregressive models gain from coarse-to-fine generation (Tian et al., 2024).

**From heuristics to learning.** Existing anisotropy is often hand-crafted (e.g., temporal correlation in video; frequency-biased processing in image pipelines), and the space of possible $M_t$ is huge, making manual search impractical. In parallel, work on *isotropic* models shows that optimizing only the discretization schedule can already boost few-step quality (Sabour et al., 2024). These trends motivate a *learned*, general-purpose anisotropic framework.

**This paper.** We introduce a trajectory-level objective that (i) trains the score network and (ii) *learns* an anisotropic schedule $M_t(\theta)$. Separately, we learn a scalar time reparameterization $r(t; \gamma)$ that reduces discretization error; both compose with a second-order sampler, yielding a practical training/inference recipe.

**Notation.** The isotropic variance-exploding (VE) process is

$$x_0 \sim p_0, \quad dx_t = dB_t \iff dx_t = -\tfrac{1}{2} \nabla \log p_t(x_t)\, dt, \tag{1}$$

with $p_t = p_0 * \mathcal{N}(0, tI)$. We generalize the above to anisotropic diffusion by letting $x_t \sim p_0 * \mathcal{N}(0, M_t)$ with a nondecreasing PSD trajectory $M_t \in \mathbb{R}^{d \times d}$:

$$x_0 \sim p_0, \quad dx_t = (\partial_t M_t)^{1/2}\, dB_t \iff dx_t = -\tfrac{1}{2}\, \partial_t M_t\, \nabla \log p_t(x_t)\, dt, \tag{2}$$

where $M_0 = 0$, $M_T = T$ for some maximum noise level $T$, and $t > s \Rightarrow M_t \succeq M_s$. We discuss further details Section 2.

---

[1] anonymous.4open.science/r/anisotropic-diffusion-paper-8738

## 1.1 MAIN CONTRIBUTIONS

We study *anisotropic* diffusion for generative modeling by learning a matrix-valued noise schedule $M_t$ that allocates noise and denoising effort across subspaces. Our contributions are:

1. **Anisotropic diffusion and reverse ODE.** We formalize variance–exploding and variance–preserving anisotropic diffusion processes, derive the corresponding reverse ODE, and give practical samplers: a first–order Euler update (7) and a *second–order Heun update* (17) and Lemma 6. This yields stable, efficient generation for the reverse anisotropic ODE.

2. **Trajectory-Level Score Matching (TLSM).** We introduce a path-integrated loss $L(\theta, \phi)$ that simultaneously (i) trains the score network to match the score and (ii) *learns* the anisotropic schedule $M_t(\theta)$ by minimizing score error along the generation trajectory (Section 3). At optimality, the network matches the exact score Lemma 1, and in the isotropic case TLSM *reduces to* weighted score matching—formally tying any choice of weights $w(t)$ to a scalar schedule $g(t)$ (Lemma 2, Section 3.1). This reveals a surprising interpretation – isotropic TLSM is equivalent to *learning an optimal weight function for score-matching.*

3. **Differentiating through $M_t(\theta)$ efficiently.** Optimizing over $M_t(\theta)$ is challenging because it involves $\partial_\theta \nabla \log p_t(x; \theta)$, which cannot be easily obtained from the score-network. We propose a *directional* estimator for $\partial_\theta \nabla \log p_t(x; \theta)$ that uses only higher-order *x-directional* derivatives of the network and is implementable in **three backward passes**, independent of $\dim(\theta)$ (Lemma 3, Section 4.1). We further present a variance-reduced formula based on estimating $\partial_\theta \left( M_t(\theta)^{-1/2} \nabla \log p_t(x; \theta) \right)$ (Lemma 5).

4. **Learning the discretization schedule.** Orthogonal to optimizing $M_t(\theta)$ wrt the *score-matching loss*, we learn a time-reparameterization $r(t; \gamma)$ that minimizes a trajectory-level *discretization error* (5). Our formulation cleanly separate score-matching from discretization-error minimization. In Algorithm 1, the learned $r(t; \gamma)$ composes with the Heun integrator based on the learned $M_t(\theta)$ noise schedule, gaining benefits from optimization of both $r(t; \theta)$ and $M_t(\theta)$.

5. **Empirical benefits.** On CIFAR-10 (Krizhevsky et al., 2009), AFHQv2 (Choi et al., 2020), and FFHQ (Karras et al., 2019), our learned anisotropic denoising model is competitive with EDM across budgets, and yields large gains on FFHQ, and at small counts e.g., **FFHQ** FID **6.02** vs. 57.14 at NFE=9 and **3.37** vs. 15.81 at NFE=13; **CIFAR-10 2.93** vs. 6.69 at NFE=13 (50k samples), with a small gap at very large NFE on CIFAR-10 (Table 1).

## 1.2 RELATED WORK

**Optimizing schedules in isotropic diffusion.** Recent work tunes the *test-time* discretization schedule to improve few-step sampling (Sabour et al., 2024; Wang et al., 2023; Liu et al., 2023; Park et al., 2024; Williams et al., 2024), complementing hand-crafted EDM designs (Karras et al., 2022). Related efforts adjust *training-time* noise weighting or sampling over noise levels while retaining a scalar schedule (Hang et al., 2024; Okada et al., 2024).

**Beyond isotropy: correlated noising.** Methods introduce structure via edge-aware anisotropy (Vandersanden et al., 2024), per-pixel multivariate schedules (Sahoo et al., 2024), or time-varying correlated masks (Huang et al., 2024). Frequency-/subspace formulations restrict or bias diffusion dynamics (Jing et al., 2022), and video models exploit structured noise across time through decomposition or temporally correlated priors (Luo et al., 2023; Ge et al., 2023; Chang et al., 2025). In contrast, we *learn* a general matrix-valued trajectory $M_t(\theta)$ together with a scalar time-transform $r(t; \gamma)$ under a trajectory-level objective, and compose both within a second-order anisotropic sampler (Algorithm 1).

## 2 PRELIMINARIES

### 2.1 ANISOTROPIC DIFFUSION: PROCESS, SCORE, AND PARAMETERIZATIONS

Recall the anisotropic diffusion process in (2). Let $M_t(\theta)$ denote the noise covariance at time $t$, parameterized by $\theta \in \mathbb{R}^c$. $p_t$ as defined in (2) has score given by

$$\nabla \log p_t(x; \theta) = M_t^{-1}(\theta) \mathbb{E}_{x_0|x_t=x} [x_0 - x_t],$$ (3)

where $(x_0, x_t)$ are defined by the joint distribution $x_0 \sim p_0$ and $x_t = x_0 + \mathcal{N}(0, M_t)$. We provide a short proof in Lemma 7 in Appendix B. In case of time-uniform isotropic diffusion (i.e. standard Brownian Motion), $M_t(\theta) = tI$, and the formula in (3) reduces to the standard score expression.

We parameterize a neural network $\texttt{net}(x, t, \phi)$ to approximate the score, we also define $\texttt{flow}$, a transformation of $\texttt{net}$ whose norm is approximately time-invariant:

$$\texttt{net}(x, t, \phi) \approx \nabla \log p_t(x; \theta), \qquad \texttt{flow}(x, t, \phi) := M_t^{1/2} \texttt{net}(x, t, \phi). \tag{4}$$

**Remark 1** (Anisotropic score matching for fixed $M_t$; not used in this paper). For a fixed $M_t$ schedule, the natural per-time objective at time $t$ is

$$\ell_t(\phi) := \mathbb{E}_{x_0, \epsilon} \left[ \left\| \texttt{net}(x_t, t, \phi) - M_t^{-1}(\theta)(x_0 - x_t) \right\|_2^2 \right], \qquad x_t = x_0 + M_t^{1/2} \epsilon. \tag{5}$$

We show in Lemma 4 in Appendix B that $\ell_t(\phi)$ is minimized by $\texttt{net}(x_t, t, \phi) = \nabla \log p_t(x)$.

**Continuous and discrete Reverse ODE for anisotropic denoising.** Given $\texttt{net}$ in (6), we define the continuous-time *forward* ODE and *reverse* ODE are respectively defined as

$$d\bar{x}_t = -\frac{1}{2} \partial_t M_t(\theta) \texttt{net}(\bar{x}_t, t, \phi) dt, \qquad \Leftrightarrow \qquad d\bar{x}_{T-t} = \frac{1}{2} \partial_t M_t(\theta) \texttt{net}(\bar{x}_{T-t}, T - t, \phi) dt. \tag{6}$$

The reverse-ODE above can be implemented via a time-discretization of (6). For intuition, we present below the simple Euler-discretization of (6): Let $K$ be number of steps, let $t_0 < t_1 ... < t_K \in [0, T]$ denote discretization points. The Euler reverse ODE is

$$x_{t_{k-1}}^{Eul} = x_{t_k}^{Eul} + (M_{t_{k-1}}^{1/2} - M_{t_k}^{1/2}) \texttt{flow}(x_{t_k}^{Eul}, t_k). \tag{7}$$

Our experiments use Heun's second order integrator (Ascher & Petzold, 1998; Karras et al., 2022) which consistently gives better FID per NFE. We detail this algorithm in Section 5.

**Variance-Preserving Anisotropic Diffusion.** It is often more useful to consider the *variance preserving* anisotropic diffusion, which is simply (time-dependent) linear-transformation of (2). Define

$$x_t^{VP} := (I + M_t(\theta))^{-1/2} x_t. \tag{8}$$

The choice of $I$ above is based on the assumption that $Cov(x_0) \approx I$. The dynamics of $x_t^{VP}$ can be explicitly written, without reference to $x_t$, using a matrix exponential. However, it is much simpler mathematically and programmatically to maintain $x_t$ explicitly, and define $x_t^{VP}$ via (8).

## 2.2 IMPLEMENTATION DETAILS: $M_t(\theta)$ FOR DCT BASIS ON IMAGES

We present below a simple example of $M_t(\theta)$ based on the 2D Discrete Cosine Transform (2D-DCT). See Appendix A background on 2D-DCT bases. Let $d = H \times H$ denote the dimension of an image. Let $v_1...v_{H^2}$ denote the 2D-DCT basis vectors of $H \times H$. Let $S_1...S_J \subset \{v_1...v_{H^2}\}$ be a disjoint union of these $H^2$ 2D-DCT vectors. For each $i = 1...J$, let $g_i(t; \theta) : \mathbb{R}^+ \to \mathbb{R}^+$ denote a monotonically increasing function satisfying $g_i(0; \theta) = 0$ and $g_i(T; \theta) = T$. Let $V_i \in \mathbb{R}^{|S_i| \times H^2}$ denote the basis matrix for $S_i$, so that $V_i^\top V_i$ is a projection matrix onto $span(S_i)$. Then we define

$$M_t(\theta) := \sum_{i=1}^{J} g_i(t; \theta) V_i^\top V_i, \qquad \text{equiv.} \qquad \partial_t M_t(\theta) := \sum_{i=1}^{J} \partial_t g_i(t; \theta) V_i^\top V_i. \tag{9}$$

We verify that $M_t(\theta) \succ M_s(\theta)$ for $t > s$, and thus defines a valid forward anisotropic diffusion process (2). Intuitively, each $g_i(t; \theta)$ defines a separate time-schedule on each subspace $S_1...S_J$.

**Efficient matrix algebra.** The form (9) implies $F(M_t) = \sum_i F(g_i(t)) V_i^\top V_i$ for $F \in \{(\cdot)^c, \partial_t, \partial_\theta\}$; our experiments use $J = 2$ and implement $g_i$ using log-linear knots (App. C).

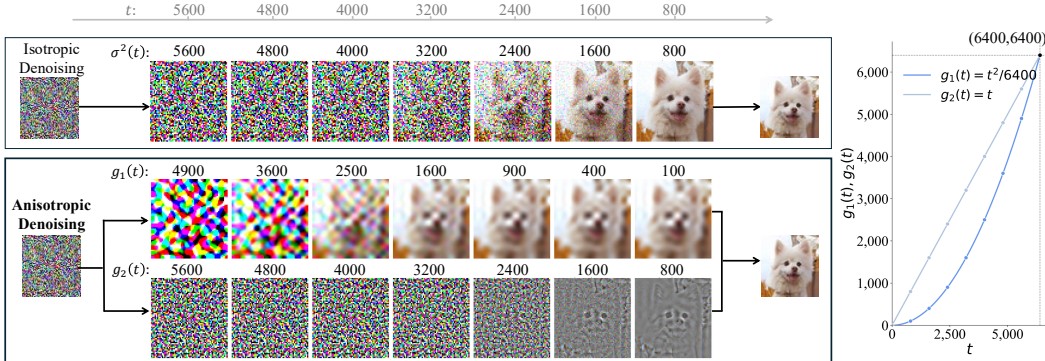

Figure 1: **Illustration of Isotropic vs. anisotropic denoising.** Top: standard isotropic sampler denoises all directions uniformly. Bottom: anisotropic sampler with two DCT subspaces, $V_1$ (low frequency) and $V_2$ (high frequency), (Section 2.2). Columns show intermediate reconstructions as $t$ decreases. The plot (right) displays the learned subspace schedules $g_1(t)$ and $g_2(t)$; the former is denoised more aggressively, thus low-frequency structure emerges earlier from the $V_1$, while high-frequency details emerge later from $V_2$. Illustration only: in practice anisotropic and isotropic will reconstruct different images, and the gap between $g_1$ and $g_2$ is typically smaller (see Fig. 3)

## 3 TRAJECTORY-LEVEL SCORE MATCHING LOSS

**Goal.** We want to learn an anisotropic noise path $M_t(\theta)$ that reduces generation error. Two error sources dominate at test time: (i) **score approximation error**, and (ii) **discretization error** of the reverse ODE. In this section we focus on (i), introducing a trajectory-level objective that jointly trains the score network and learns $M_t(\theta)$. We discuss (ii) in Section 5.

From (2) and (6), the *variance-preserving* ODE (8) with $\nabla \log p_t$ and net are defined by the drift velocity fields $v(x_t, t; \theta)$ and $\bar{v}(\bar{x}_t, t; \theta)$ respectively:

$$v(x, t; \theta) := -(I + M_t(\theta))^{-1/2} \partial_t M_t(\theta) \nabla \log p_t(x; \theta) - \frac{1}{2}(I + M_t(\theta))^{-3/2} \partial_t M_t(\theta) x,$$

$$\bar{v}(x, t; \theta, \phi) := -(I + M_t(\theta))^{-1/2} \partial_t M_t(\theta) \mathrm{net}(x, t, \phi) - \frac{1}{2}(I + M_t(\theta))^{-3/2} \partial_t M_t(\theta) x. \quad (10)$$

Let us also define $\tilde{v}(x, y, t; \theta) := -(I + M_t(\theta))^{-1/2} \partial_t M_t(\theta) M_t^{-1}(\theta)(y - x) - \frac{1}{2}(I + M_t(\theta))^{-3/2} \partial_t M_t(\theta) x$. It follows from (3) that $v(x, t; \theta) = \mathbb{E}_{x_0|x_t=x}[\tilde{v}(x, x_0, t; \theta)]$. For $\epsilon \sim \mathcal{N}(0, I)$, $x_t := x_0 + M_t^{1/2}(\theta)\epsilon$, we now define the trajectory-level score-matching loss as

$$L(\theta, \phi) = \int_0^T \mathbb{E}_{x_0, \epsilon} \left[ \|\bar{v}(x_t, t; \theta, \phi) - \tilde{v}(x_t, x_0, t; \theta)\|_2^2 \right] dt, \quad (11)$$

where $T$ denotes maximum noise level. We also provide a more explicit expression of $L(\theta, \phi)$ in (15) in Section 4.2 below. $L(\theta, \phi)$ can be viewed as a generalization to the standard score-matching objective, but with *matrix-valued weights*. The loss in (11) has a number of desirable properties:

**Exact score at optimality.** The following analog of Lemma 4 shows that $L(\theta, \phi)$, like the standard score-matching loss, also encourages net to match the score. Proof in Appendix B.

**Lemma 1.** $L(\theta, \phi)$, as defined in (11), is minimized if $\mathrm{net}(x, t; \phi) = \nabla \log p_t(x; \theta)$ for all $(x, t)$.

**Connection to path-level KL divergence:** For two stochastic processes evolving as $dx_t = v(x_t, t)dt + dB_t$ and $d\bar{x}_t = \bar{v}(\bar{x}_t, t)dt + dB_t$, the path-level KL divergence is bounded by $\int_0^T \mathbb{E}[\|\bar{v}(x_t, t) - v(x_t, t)\|]_2^2 dt$ (assuming sufficient regularity, e.g. Novikov's condition). This has been used, for instance, to bound the discretization error of the reverse *SDE* in Chen et al. (2022). Our loss (1) differs from the KL upper bound in replacing $v(x, t; \theta)$ by $\tilde{v}(x, x_0; \theta)$, because the true score (and hence $v(x, t; \theta)$) is not accessible during training. In this paper, we focus on the forward and reverse ODE for simplicity, but the forward and reverse SDE can be analogously defined.

**Integration error under VP scaling (intuition).** We choose to compute the score-matching error under the VP formulation for two reasons: (1) VP transformation keeps scale roughly constant wrt time, and (2) at large time, the backward ODE is dominated by $x_t$ contracting towards $0$. Thus discretization errors at high noise should be discounted (via the $(I + M_t(\theta))^{-1/2}$ scaling).

## 3.1 Choice of weight $w(t)$ is equivalent to choice of noise-schedule $g_t(\theta)$.

Possibly of independent interest, we present here a connection between learning $g_t(\theta)$, and the standard score-matching formuation. Consider the *isotropic* version of (11). Let $M_t(\theta) := g_t(\theta)I$, where $g_t$ is a scalar-valued monotonically increasing function. $L(\theta, \phi)$ thus simplifies to

$$\int_0^T \frac{(\partial_t g_t(\theta))^2}{1 + g_t(\theta)} \mathbb{E}_{x_0, \xi} \left[ \left\| \mathtt{net}_{g_t(\theta)}(x_0 + g_t(\theta)^{1/2}\xi; \phi) + g_t(\theta)^{-1/2}\xi \right\|_2^2 \right] dt. \tag{12}$$

With slight abuse of notation we let $\mathtt{net}_{\sigma^2}(x_t; \phi)$ denote the network trained to match the score of $p_0 * \mathcal{N}(0, \sigma^2 I)$. In literature, the score-matching loss is usually a weighted average $\int_0^T w(s)\mathbb{E}_{x_0, \xi} \left[ \|\mathtt{net}_s(x_0 + \sqrt{s}\xi) + \xi/\sqrt{s}\|_2^2 \right] ds$. We show below that choosing a $g_t(\theta)$ is exactly equivalent to choosing a weighing function $w(s)$:

**Lemma 2.** *For any $w(t)$, there exists a $g_t(\theta)$ and constant c, such that for any $H(t)$*

$$\int_0^T \frac{(\partial_t g_t(\theta))^2}{1 + g_t(\theta)} H(g_t(\theta)) dt = c \int_0^T w(t)H(t)dt.$$

We defer the proof to Appendix B. Consequently, *any weighted score-matching loss* for isotropic diffusion (where the weights can be a combination of explicit weighting function and implicit distribution density, e.g. Karras et al. (2022)) can be equivalently written as an instance of *trajectory-level score-matching loss*, for a specific choice of $g_t(\theta)$. When we optimize over the space of noise-schedules $g_t(\theta)$ wrt $L(\theta, \phi)$, we are *equivalently optimizing over the choice of weighing function $w(t)$ under the standard score-matching loss*.

# 4 Optimization Score Matching Loss over $M_t(\theta)$

For fixed $t$, let $\theta \in \mathbb{R}^c$ be the vector parameterizing $M_t(\theta)$. Then for all $i = 1...c$,

$$\partial_{\theta_i} x_t(\theta) = -\frac{1}{2}\partial_{\theta_i} M_\theta \nabla \log p_t(x; \theta) \quad \Leftrightarrow \quad \partial_{\theta_i} p_t(x; \theta_i) = \frac{1}{2}\mathbf{div}(\partial_{\theta_i} M_t(\theta_i)\nabla p_t(x; \theta_i)). \tag{13}$$

The LHS of (13) resembles (2), as both describe the density evolution of $p_t(x; \theta)$, and follow almost identical proofs. However, do note that (2) and (13) *have very different meanings*. Specifically, (2) holds $\theta$ fixed, and evolves $x_t(\theta)$ over $t$, whereas (13) holds $t$ fixed, and evolves $x_t(\theta)$ over $\theta$.

## 4.1 Stochastic Approximation to $\partial_\theta \nabla \log p_t(x; \theta)$ and $\partial_\theta \mathrm{NET}(x, t, \phi)$

A significant challenge of optimizing $L(\theta, \phi)$ lies in the fact that *there is no simple way to approximate $\partial_\theta \nabla \log p_t(x; \theta)$*. This is because, whereas $\mathtt{net}(x, t; \phi) \approx \nabla \log p_t(x; \theta)$ is a good approximation of the *value* of the score, it does not explicitly provide the *derivative* of the score, with respect to $\phi$. One simple approach is to allow $\mathtt{net}(x, t, \phi, \theta)$ to additionally take in $\theta$ as an input argument, e.g. via a more complex time-embedding module, but this approach has two major downsides:

1. The score-matching loss (11) needs to integrate over not just time $t \in [0, T]$, but over a large set of potential $\theta$'s. This is forces the $\mathtt{net}$ to trade-off the score loss at various suboptimal $\theta$ values, which are not used for inference-time reverse-ODE.

2. As the parameterization of $M_t(\theta)$ as a function of $\theta$ becomes more complex, $\mathtt{net}(x, t, \phi, \theta)$ must also use a more complex time-embedding module to encode $(t, \theta)$.

In contrast, we present a principled approach, that computes an unbiased stochastic estimate of the $\theta$-**space derivative** $\partial_{\theta_i} \nabla \log p_t(x; \theta)$, using only **higher-order directional $x$-space derivatives** $\nabla \log p_t(x; \theta)$ **along specific directions**. Programmatically (e.g. in PyTorch), the derivatives with respect to all of $\theta_1...\theta_c$ is computed together in three backward passes, so the additional computational cost is agnostic to the dimension of $\theta$, and the parameterization of $M_t(\theta)$.

**Lemma 3.** *Let $e_1...e_d$ denote any orthonormal basis of $\mathbb{R}^d$. Then*

$$\partial_{\theta_i} \nabla \log p_t(x; \theta) = \frac{1}{2} \sum_{j=1}^{d} \partial_r \partial_s \nabla \log p_t(x + re_j + s\partial_{\theta_i} M_t(\theta)e_i; \theta)$$
$$+ \partial_s \nabla \log p_t(x + s\partial_{\theta_i} M_t(\theta) \nabla \log p_t(x; \theta); \theta).$$

Applying the approximation of $\mathtt{net}(x, t, \phi) \approx \nabla \log p_t(x; \theta)$ on both sides gives

$$\partial_{\theta_i} \mathtt{net}(x, t, \phi) \approx \frac{1}{2} \sum_{i=1}^{d} \partial_r \partial_s \mathtt{net}(x + re_i + s\partial_{\theta_i} M_t(\theta)e_i, t, \phi)$$
$$+ \partial_s \mathtt{net}(x + s\partial_{\theta_i} M_t(\theta)\mathtt{net}(x, t, \phi), t, \phi). \tag{14}$$

The sum over $j = 1...d$ is expensive to compute exactly, but it can be efficiently approximated in expectation, by sampling $e_i$ from the standard Gaussian distribution.

## 4.2 Optimizing the loss $L(\theta, \phi)$

We now apply our formula from Section 4.1 to optimize $L(\theta, \phi)$ in (11). For notational clarity, we treat $\theta$ as a scalar. Mathematically, $\theta \in \mathbb{R}^c$ can be handled by repeating the computation for each scalar $\theta_i$. At the end of this section, we provide PyTorch code, showing how the gradients of all $\theta_1...\theta_c$ can be *simultaneously computed in one set of backward passes*. Let $x_0 \sim p_0$ and $\xi \sim \mathcal{N}(0, I)$ independently, and define $x_t := x_0 + M_t^{1/2}\xi$, consistent with (2). Following the setup in Section 3, $L(\theta, \phi)$ is equal to

$$\mathbb{E}_{x_0, \xi} \left[ \left\| (I + M_t(\theta))^{-1/2} \partial_t M_t(\theta) \left( \mathtt{net}(x_0 + M_t^{1/2}(\theta)\xi, t, \phi) + M_t(\theta)^{-1/2}\xi \right) \right\|_2^2 \right]. \tag{15}$$

Let us define the gradient of $L(\theta, \phi)$ above with respect to $\mathtt{net}$ as.

$$G(\theta, \phi) := 2\mathbb{E}_{x_0, \xi} \left[ \partial_t M_t(\theta)(I + M_t(\theta))^{-1} \partial_t M_t(\theta) \left( \mathtt{net}(x_0 + M_t^{1/2}(\theta)\xi, t, \phi) + M_t(\theta)^{-1/2}\xi \right) \right].$$

To estimate the actual derivative of $L(\theta)$, accounting for the change-in-score-due-to-$\theta$, we augment the derivative $\partial_\theta(15)$ using

$$\partial_\theta L(\theta, \phi) = \partial_\theta(15) + \langle G(\theta, \phi), (14) \rangle. \tag{16}$$

The expectation wrt $x_0, \xi$ can be approximated by a finite sum over $j = 1...n$ of $\{(x_0^{(j)}, \xi^{(j)})\}_{j=1...n}$, sampled iid from $p_0 \times \mathcal{N}(0, I)$. We highlight below two aspects of practical implementation.

## 4.3 Implementation Details

**Time embedding and detaching $\theta$.** In common implementations, $\mathtt{net}(x, \sigma(t), \phi)$ takes as input the noise-level $\sigma$, and not the time index. In our actual experiment setup described in Section 2.2, we use $\mathtt{net}(x, \tilde{\sigma}(t; \theta), \phi)$, with $\tilde{\sigma}(t; \theta) := \sqrt{g_1(t; \theta)g_2(t; \theta)}$ to replace $\sigma(t)$ as this requires minimal retraining of the time-embedding of the original net (and requires no retraining if $g_1 = g_2$). In the implementation, it is important to detach the $\theta$ from the computation graph of $\tilde{\theta}$, so as not to double-count the derivative wrt $\theta$. We also emphasize that backpropagating through $\tilde{\sigma}(t; \theta)$ is insufficient for estimating the derivative $\partial_\theta \mathtt{net}$, because $\tilde{\sigma}$ is a scalar-valued "projection" of the full matrix-valued $M_t(\theta)$ noise, and thus we still need to use (14).

**Variance Reduction with $\partial_\theta \mathtt{flow}$ instead of $\partial_\theta \mathtt{net}$.** The scale of $\|\mathtt{net}(x, t; \phi)\|_2 \approx \|\nabla \log p_t(x; \theta)\|_2 \approx \|M_t^{-1/2}(\theta)\|_2$ can vary significantly with the noise level. This could lead to high variance in the stochastic-estimation of $\partial_\theta \mathtt{net}$ in (14). To address this, we propose a mathematically equivalent estimate of $\partial_\theta \mathtt{net}$ based on $\mathtt{flow}(x, t, \phi) := M_t^{1/2}(\theta)\mathtt{net}(x, t, \phi)$, whose scale is approximately constant across time: $\|\mathtt{flow}(x, t, \phi)\|_2 \approx \|M_t^{1/2} \nabla \log p_t(x; \theta)\|_2 \approx d$.

(`flow` is defined in (4)) We show in Lemma 5 that

$$\partial_\theta \texttt{flow}(x,t,\phi) = \frac{1}{2}\sum_{i=1}^{d}\partial_r\partial_s\texttt{flow}(x+re_i+s\partial_\theta M_t(\theta)e_i,t,\phi)$$

$$+ \partial_s\texttt{flow}(x+s\partial_\theta M_t^{1/2}(\theta)\texttt{flow}(x,t,\phi),t,\phi) + \frac{1}{2}M_t^{-1}(\theta)(\partial_\theta M_t(\theta))\texttt{flow}(x,t,\phi)$$

$$H(\theta,\phi) = 2\mathbb{E}_{x_0,\xi}\left[M_t(\theta)^{-1/2}\partial_t M_t(\theta)(I+M_t(\theta))^{-1}\partial_t M_t(\theta)M_t(\theta)^{-1/2}\Big(\texttt{flow}(x_0+M_t^{1/2}(\theta)\xi,t,\phi)+\xi\Big)\right]$$

$$\partial_\theta L(\theta,\phi) = \partial_\theta(15) + \langle H(\theta,\phi),\partial_\theta\texttt{flow}(x,t,\phi)\rangle.$$

The last line above is equivalent to (16), but written with `flow` instead of `net`.

## 5  LEARNING SCALAR DISCRETIZATION SCHEDULE

In this section, we present the implementation of Heun's second-order backward ODE integrator for anisotropic diffusion. Additionally, we discuss a way to select an optimal denoising schedule $r(t;\gamma) : [0,T] \to [0,T]$ based on a *trajectory-level discretization loss*. We emphasize that the optimal denoising schedule $r(t;\gamma)$ can be composed with the optimal score-matching noise schedule $M_t(\theta)$ obtained from minimizing $L(\theta,\phi)$ in (11). In this section, we omid dependence on $\theta,\phi$.

### 5.1  HEUN'S SECOND-ORDER ALGORITHM FOR ANISOTROPIC DIFFUSION DENOISING.

Let $\tilde{u}$ be the estimate of $\texttt{flow}(\bar{x}_t,\bar{t})$, given two evaluations of `flow` at $(x,t)$ and $\hat{x},\hat{t}$ respectively:

$$\tilde{u}(\bar{t};x,\hat{x},t,\hat{t}) := \texttt{flow}(x,t) + (M_{\bar{t}}^{1/2}-M_t^{1/2})(M_{\hat{t}}^{1/2}-M_t^{1/2})^{-1}\big(\texttt{flow}(\hat{x},\hat{t})-\texttt{flow}(x,t)\big).$$

Let $t_0 < t_1 < ... < t_K$ denote $K$ discretization points with $t_0 = 0$ and $t_K = K$. Let $t_{k-1} \le \hat{t}_k < t_k$ denote a set of secondary evaluation points. Then Heun's second-order backward ODE is defined as

$$\hat{x}_{\hat{t}_k} = \tilde{x}_{t_k} + (M_{\hat{t}_k}^{1/2}-M_{t_k}^{1/2})\texttt{flow}(\tilde{x}_{t_k},t_k),$$

$$\tilde{x}_{t_{k-1}} = \tilde{x}_{t_k} + \int_{t_k}^{t_{k-1}}(\partial_t M_{\bar{t}}^{1/2})\tilde{u}(\bar{t};\tilde{x}_{t_k},\hat{x}_{\hat{t}_k},t_k,\hat{t}_k)d\bar{t}. \tag{17}$$

We verify in Lemma 6 that $\int_{t_k}^{t_{k-1}}\tilde{u}(\bar{t};\tilde{x}_{t_k},\tilde{x}_{\hat{t}_k},t_k,\hat{t}_k)d\bar{t}$ has a simple closed-form expression:

$$(M_{t_{k-1}}^{1/2}-M_{t_k}^{1/2})(\texttt{flow}(\tilde{x}_{t_k},t_k))-\frac{1}{2}(M_{t_{k-1}}^{1/2}-M_{t_k}^{1/2})^2(M_{\hat{t}_k}^{1/2}-M_{t_k}^{1/2})^{-1}\big(\texttt{flow}(\hat{x}_{\hat{t}_k},\hat{t}_k)-\texttt{flow}(\tilde{x}_{t_k},t_k)\big).$$

In general, the choices of evaluation points $t_k$ and $\hat{t}_k$ can have a significant effect on the discretization error. In Karras et al. (2022), for isotropic diffusion models, the authors choose a schedule which corresponds to $t_k \approx \left(\sigma_{\max}^{1/\rho} - \frac{K-k}{K}\left(\sigma_{\min}^{1/\rho}-\sigma_{\max}^{1/\rho}\right)\right)^\rho$, with $\rho = 7$ being an empirically chosen hyperparameter, and $\sigma_{\min} \approx 0, \sigma_{\max} \approx T, \hat{t}_k = t_{k-1}$. In the next section, we present a principled way to select a discretization schedule by minimizing the trajectory-level discretization error.

### 5.2  OPTIMAL DISCRETIZATION SCHEDULE

We will let $r(t;\gamma) : [0,T] \to [0,T]$ denote a monotonically increasing time-transformation with $r(0;\gamma) = 0, r(T;\gamma) = T$. We will optimize over the choice of discretization schedules $r(t;\gamma)$. Let $x_t$ denote the continuous-time backward ODE, defined as the time-reversal of

$$dx_t = -\frac{1}{2}\partial_t M_t\texttt{net}(x_t,t) = -\partial_t M_t^{1/2}\texttt{flow}(x_t,t). \tag{18}$$

On the other hand, (17) is equivalent to $d\tilde{x}_t = -(\partial_t M_t^{1/2})\tilde{u}(\bar{t};\tilde{x}_{t_k},\hat{x}_{\hat{t}_k},t_k,\hat{t}_k)$ for $t \in [t_{k-1},t_k]$. Again inspired by the Girsanov's Theorem, which gave rise to our *trajectory-level score-matching loss* $L(\theta,\phi)$, we define the *idealized trajectory-level discretization loss* $\hat{H}(\gamma)$ as

$$\hat{H}(\gamma) = \int_0^T \mathbb{E}\left[\left\|\partial_t M_{r(t;\gamma)}^{1/2}\Big(\texttt{flow}(x_{r(t;\gamma)},r(t;\gamma)) - \tilde{u}(r(t;\gamma);\tilde{x}_{r(t_k;\gamma)},\hat{x}_{r(\hat{t}_k;\gamma)},r(t_k;\gamma),r(\hat{t}_k;\gamma))\Big)\right\|_2^2\right]dt,$$

where in the above, $t_{k-1}, t_k$ are the two evaluation points such that $r(t; \gamma) \in [r(t_{k-1}; \gamma), r(t_k; \gamma)]$. In practice, we optimize a stochastic approximation of $\hat{H}$ defined by

$$H(\gamma) = \mathbb{E}_{t,\tilde{t},\tilde{x},\hat{x}} \left[ \left\| \partial_t M_{r(t;\gamma)}^{1/2} \big( \text{flow}(x_{r(t;\gamma)}, r(t;\gamma)) - \tilde{u}(r(t;\gamma); \tilde{x}, \hat{x}, r(\tilde{t};\gamma), r(\hat{t};\gamma)) \big) \right\|_2^2 \right] dt,$$

where $\tilde{t} \sim Unif([0, T])$, $\bar{t} = \max\{0, \tilde{t} - T/8\}$, $t \sim Unif([\bar{t}, \tilde{t}])$. $\hat{t} = (\bar{t} + \hat{t})/2$. In addition to the discretization error, recall the continuous-time score-matching loss $L$ defined in (11). For $\theta, \phi$ fixed, and under the time-transformation $r(t; \gamma)$, the score-matching loss is given by

$$\tilde{L}(\gamma) := \int_0^T \mathbb{E}_{x_0, \epsilon} \left[ \left\| \bar{v}(x_{r(t;\gamma)}, r(t;\gamma); \theta, \phi) - \tilde{v}(x_{r(t;\gamma)}, x_0, r(t;\gamma); \theta) \right\|_2^2 \right] dt, \qquad (19)$$

where $\bar{v}$ and $\tilde{v}$ are as defined in Section 3, and $x_{r(t;\gamma)} = x_0 + M_{r(t;\gamma)}^{1/2} \epsilon$. Combining the above, we optimize $r(\cdot; \gamma)$, over the space of $\gamma$'s, to minimize $H(\gamma) + \tilde{L}(\gamma)$. We parameterize $r(t; \gamma)$ the same as $g_i(t; \theta)$ (Section 2.2). In practice, we replace $\tilde{L}$ by $\hat{L}$, which is a more elaborate version of $\tilde{L}$ that is a more accurate estimator of the score-matching loss (see (20) in Appendix D). The following algorithm combines all our previous optimization techniques:

---

**Algorithm 1** Combining all training

1: Train $(\theta^*, \phi^*)$ on loss $L(\theta, \phi)$
2: Given $M_t(\theta^*)$ and $\text{net}(\cdot, \cdot, \phi^*)$, train $\gamma^*$ as described in Section 5.2.
3: Let $s_i = iT/K$ denote a *uniform grid*. Let $t_i = r(s_i, \gamma^*)$ for $i = 0...K$. Let $\hat{t}_i = r((s_{i-1} + s_i)/2, \gamma^*)$.
4: To generate samples, implement (17), with $t_i$ and $\hat{t}_i$ from step 3 above.

---

## 6 EXPERIMENTAL EVALUATION

We evaluate our anisotropic diffusion schedules on three standard image generation benchmarks: CIFAR-10 ($32 \times 32$) (Krizhevsky et al., 2009), AFHQv2 ($64 \times 64$) (Choi et al., 2020), and FFHQ ($64 \times 64$) (Karras et al., 2019). All experiments are compared against the EDM baseline (Karras et al., 2022), using the official generation code and their best-reported settings. Our models are finetuned from the corresponding EDM networks, consuming the equivalent of 1.2M image passes over the course of training. For evaluation, we generate 50k samples and compute the Fréchet Inception Distance (FID↓). Results are reported across a range of function evaluations (NFE), following the same experimental settings as Karras et al. (2022). No additional hyperparameters are tuned.

**Algorithm details.** (1) EDM is EDM baseline. (2) $g^{iso}$ parameterizes an isotropic noise schedule ($M_t$ with $J = 1$ in (9)). $(\theta, \phi)$ is trained on $L(\theta, \phi)$ and generation uses (17), with uniform grid $t_k$ and $\hat{t}_k = (t_{k-1} + t_k)/2$. $(g_1^{ani}, g_2^{ani})$ parameterize $M_t$ with $J = 2$ in (9); the training/inference procedure is identical to $g^{iso}$. $g_w^{iso}$ (resp $(g_{1,w}^{ani}, g_{2,w}^{ani})$) is generated using Algorithm 1, and parameterizes $M_t$ with $J = 1$ (resp $J = 2$). For the $J = 2$ setups, we choose $V_1$ to contain the $H^2/4$ lowest-frequency DCT bases, and $V_2$ to contain the remainder bases, where $H$ is the image resolution (e.g., $H = 64$ for $64 \times 64$). $g_1$ and $g_2$ are their respective schedules.

**Comparable overall performance.** Across datasets, our learned schedules achieve performance broadly comparable to EDM. As shown in Table 1, the reported FIDs remain close to those of the baseline over a wide range of NFE. The only noticeable deviation occurs on CIFAR-10 at large NFE, where performance is slightly worse, but the gap is minor relative to the overall trend.

**Significant gains at low NFE.** Our methods show consistent advantages over EDM in the low-NFE regime, often by a large margin (Table 1). On CIFAR-10, $(g_1^{ani}, g_2^{ani})$ achieves FID=2.93 at NFE=13. On AFHQv2, $(g_{1,w}^{ani}, g_{1,w}^{ani})$ achieves FID=2.42 at NFE=19. On FFHQ, the same variant reaches FID=3.37 at NFE=13.

**Strong improvements on FFHQ.** The largest gains are observed on FFHQ, a more complex human-face dataset. Across all NFE values, learned schedules outperform EDM. At smaller NFE (e.g., 9–13 steps), the improvements are dramatic: at NFE=9, our method achieves FID=6.02 compared to 57.14 for EDM; at NFE=11, 4.25 vs 29.39; and at NFE=13, 3.37 vs 15.81.

| **CIFAR-10** | | | | | | | | | |
|---|---|---|---|---|---|---|---|---|---|
| Method | nfe 9 | nfe 11 | nfe 13 | nfe 15 | nfe 17 | nfe 35 | nfe 59 | nfe 79 | Best FID |
| EDM | 35.52 | 14.37 | 6.694 | 4.231 | 3.027 | **1.829** | **1.868** | **1.890** | 1.829 |
| $g^{iso}$ | 49.06 | 31.17 | 19.53 | 13.03 | 9.082 | 2.134 | 1.928 | 1.946 | 1.928 |
| $g_w^{iso}$ | 5.133 | 14.05 | 6.898 | **2.633** | 2.585 | 2.003 | 1.955 | 1.949 | 1.949 |
| $(g_1^{ani}, g_2^{ani})$ | 5.849 | **3.567** | **2.927** | 2.638 | **2.469** | 2.128 | 2.091 | 2.054 | 2.054 |
| $(g_{1,w}^{ani}, g_{2,w}^{ani})$ | **4.672** | 6.060 | 5.689 | 2.759 | 2.536 | 2.078 | 2.082 | 2.039 | 2.039 |
| **AFHQv2** | | | | | | | | | |
| Method | nfe 9 | nfe 11 | nfe 13 | nfe 15 | nfe 19 | nfe 39 | nfe 79 | nfe 119 | Best FID |
| EDM | 27.98 | 13.66 | 7.587 | 4.746 | 2.986 | 2.075 | 2.042 | 2.046 | 2.042 |
| $g^{iso}$ | 35.38 | 15.20 | 10.33 | 8.287 | 5.684 | 2.332 | 2.103 | 2.068 | 2.068 |
| $g_w^{iso}$ | 4.745 | 3.766 | 2.920 | 2.564 | 2.495 | 2.123 | 2.088 | 2.067 | 2.067 |
| $(g_1^{ani}, g_2^{ani})$ | 22.20 | 11.67 | 8.498 | 6.755 | 4.406 | 2.167 | 2.039 | 2.023 | 2.023 |
| $(g_{1,w}^{ani}, g_{2,w}^{ani})$ | **4.697** | **3.590** | **2.859** | **2.445** | **2.416** | **2.061** | **2.036** | **2.023** | 2.023 |
| **FFHQ** | | | | | | | | | |
| Method | nfe 9 | nfe 11 | nfe 13 | nfe 15 | nfe 19 | nfe 39 | nfe 79 | nfe 119 | Best FID |
| EDM | 57.14 | 29.39 | 15.81 | 9.769 | 5.169 | 2.575 | 2.391 | 2.374 | 2.374 |
| $g^{iso}$ | 68.35 | 27.92 | 13.44 | 8.097 | 3.958 | **2.265** | **2.242** | **2.281** | 2.242 |
| $g_w^{iso}$ | 6.679 | 4.290 | 3.872 | 3.472 | 3.033 | 2.365 | 2.281 | 2.292 | 2.281 |
| $(g_1^{ani}, g_2^{ani})$ | 45.43 | 17.21 | 8.263 | 5.129 | 3.001 | 2.327 | 2.313 | 2.354 | 2.313 |
| $(g_{1,w}^{ani}, g_{2,w}^{ani})$ | **6.016** | **4.253** | **3.365** | **3.119** | **2.829** | 2.359 | 2.309 | 2.348 | 2.309 |

Table 1: FID ↓ vs. NFE across datasets (50k samples). For each method, we perform 3 independent random generations of 50k images and report the minimum FID across the three runs. *Bold = per-NFE best. Blue = Best FID lower than EDM.*

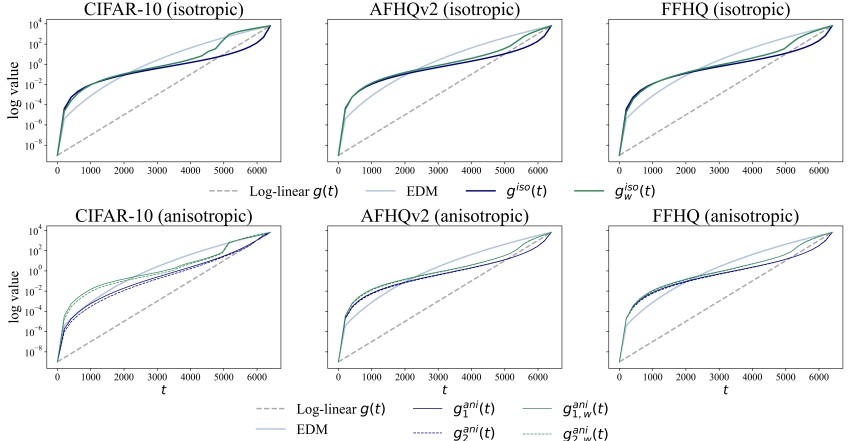

Figure 2: Learned schedules for isotropic and anisotropic cases.

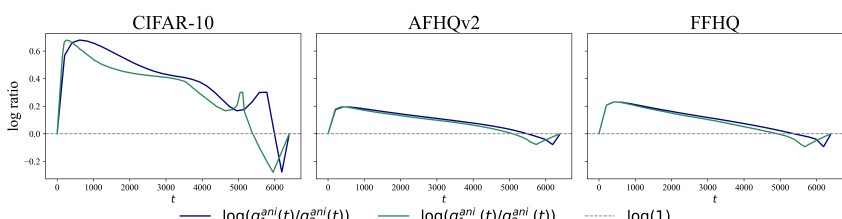

Figure 3: $\log(g_1^{ani}(t)/g_2^{ani}(t))$ and $\log(g_{1,w}^{ani}(t)/g_{2,w}^{ani}(t))$.

## REPRODUCIBILITY STATEMENT

The code used to run all experiments is linked in the abstract. We provide detailed descriptions of datasets, architectures, training settings, and evaluation protocols. Every theorem or lemma stated or referenced in the main text is accompanied by a complete proof, either in the main body or in the Appendix.

## ETHICS STATEMENT

We rely exclusively on publicly available datasets (CIFAR-10, AFHQv2, FFHQ), which are widely used in the machine learning community and distributed for research purposes. Our work is methodological in nature, with experiments confined to standard benchmarks. We do not anticipate any significant ethical risks arising from this study.

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

## A  2D-DCT TRANSFORM

Let $H$ be the image side length and $d = H^2$. The two-dimensional DCT (type-II) basis over $\mathbb{R}^{H \times H}$ is defined as follows. For each pair $(p, q) \in \{0, \ldots, H-1\}^2$, the basis is

$$\Phi_{p,q}(x, y) = \gamma_p \gamma_q \, \cos\left(\tfrac{(2x+1)p\pi}{2H}\right) \cos\left(\tfrac{(2y+1)q\pi}{2H}\right), \qquad x, y = 0, \ldots, H-1,$$

with normalization factors

$$\gamma_p = \begin{cases} H^{-1/2}, & p = 0, \\ \sqrt{2}\, H^{-1/2}, & p > 0, \end{cases} \qquad \gamma_q = \begin{cases} H^{-1/2}, & q = 0, \\ \sqrt{2}\, H^{-1/2}, & q > 0. \end{cases}$$

Vectorizing each $\Phi_{p,q}$ into $\mathbb{R}^d$ and enumerating them yields the orthonormal basis $\{v_1, \ldots, v_{H^2}\}$, which are the 2D-DCT basis of $\mathbb{R}^d$. Please refer to Figure 4 for example 2D-DCT bases.

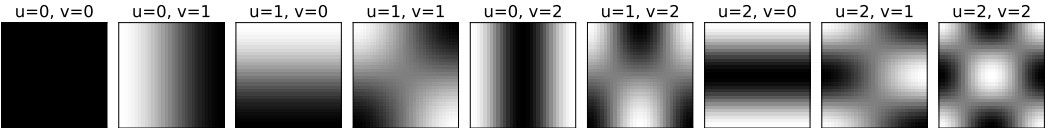

Figure 4: The first nine 2D-DCT bases ordered by increasing frequency.

## B  PROOFS

**Lemma 4.** *Let $\ell_t(\phi)$ be as defined in (5). Then $\ell_t(\phi)$ is minimized if $\texttt{net}(x, t; \phi) = \nabla \log p_t(x; \theta)$.*

*Proof of Lemma 4.* To see this, let $V : \mathbb{R}^d \to \mathbb{R}^d$ be an arbitrary vector field. With abuse of notation, define

$$\ell_t(V) := \mathbb{E}_{x_0, \epsilon} \left[ \left\| V(x_t) - M_t^{-1}(\theta)(x_0 - x_t) \right\|_2^2 \right].$$

Observe that $\ell_t$ above can be minimized pointwise at each $x_t$. By law of iterated expectation, $\mathbb{E}_{x_0, \epsilon} \left[ \left\| V(x_t) - M^{-1}(x_0 - x_t) \right\|_2^2 \right] = \mathbb{E}_{x_t} \left[ \mathbb{E}_{x_0, \epsilon | x_t} \left[ \left\| V(x_t) - M^{-1}(x_0 - x_t) \right\|_2^2 \right] \right]$. Further observe that

$$\arg \min_{v \in \mathbb{R}^d} \mathbb{E}_{x_0, \epsilon | x_t} \left[ \left\| v - M_t^{-1}(\theta)(x_0 - x_t) \right\|_2^2 \right] = M_t^{-1}(\theta) \mathbb{E}_{x_0, \epsilon | x_t} [x_0 - x_t] = \nabla \log p_t(x; \theta).$$

The last equality follows from (3). □

*Proof of Lemma 1.* We simplify the term inside the Euclidean norm in (11):

$$\bar{v}(x_t, t; \theta, \phi) - \tilde{v}(x_t, x_0, t; \theta) = (I + M_t(\theta))^{-1/2} \partial_t M_t(\theta) (M_t^{-1}(\theta)(x_0 - x_t) - \texttt{net}(x_t, t, \phi)).$$

Let $M_{\text{all}}$ denote $(I + M_t(\theta))^{-1/2} \partial_t M_t(\theta)$.

Following Lemma 4, let $V : \mathbb{R}^d \to \mathbb{R}^d$ be an arbitrary vector field.

We can minimize the expectation pointwise at each $x_t$. We can rewrite the expectation

$$\mathbb{E}_{x_0, \epsilon} \left[ \left\| M_{\text{all}}(\texttt{net}(x_t, t, \phi) - M_t^{-1}(\theta)(x_0 - x_t)) \right\|_2^2 \right]$$

as

$$\mathbb{E}_{x_0, \epsilon} \left[ \left\| M_{\text{all}}(V(x_t) - M_t^{-1}(\theta)(x_0 - x_t)) \right\|_2^2 \right] = \mathbb{E}_{x_t} \left[ \mathbb{E}_{x_0, \epsilon | x_t} \left[ \left\| M_{\text{all}}(V(x_t) - M_t^{-1}(\theta)(x_0 - x_t)) \right\|_2^2 \right] \right]$$

by the law of iterated expectation.

$$\arg \min_{v \in \mathbb{R}^d} \mathbb{E}_{x_0, \epsilon | x_t} \left[ (v - M_t^{-1}(\theta)(x_0 - x_t))^\top M_{\text{all}} (v - M_t^{-1}(\theta)(x_0 - x_t)) \right]$$

$$= M_t^{-1}(\theta) \mathbb{E}_{x_0, \epsilon | x_t} [x_0 - x_t] = \nabla \log p_t(x; \theta).$$

The above equality follows from (3). The penultimate equality follows from the fact that $\arg \min_{a \in \mathbb{R}^d} (b - a)^\top Q(b - a) = b$ for any PSD matrix $Q$. Note that $\partial_t M_t = A_t^2$ is PSD, and $M_t$ being the covariance in the diffusion process is also PSD. Hence, $M_{\text{all}}$ is PSD. □

*Proof of Lemma 2.* Let $w : [0,T] \to (0,\infty)$ be measurable and define

$$\Phi(x) := \int_0^x \frac{ds}{(1+s)\,w(s)} < \infty.$$

Then, to prove this Lemma, it is sufficient to show there exist a $c = \frac{\Phi(T)}{T} > 0$ and a strictly increasing continuous function $g : [0,T] \to [0,T]$ with $g(0) = 0$, $g(T) = T$ such that

$$\frac{g'\big(g^{-1}(t)\big)}{1+t} = c\,w(t) \text{ for all } t.$$

where $c = \Phi(T)/T$.

Define $g$ implicitly by $\Phi\big(g(t)\big) = ct$ for $(0 \le t \le T)$, i.e. $g(t) = \Phi^{-1}(ct)$. Differentiating $\Phi(g(t)) = ct$ with respect to $t$ yields the separable ODE $g'(t) = (1 + g(t))\,c\,w(g(t))$, and with $r = g(t)$ this is $\frac{g'(g^{-1}(r))}{1+r} = c\,w(r)$. Hence, we derive an expression for $g$ involving a constant $c$ and any $w$. Monotonicity of $g(t)$ follows since $w > 0$ and $c > 0$ imply $g'(t) > 0$.

Substituting into $c\int_0^T w(t)H(t)dt$, we get

$$c\int_0^T w(r)H(r)dr = \int_0^T \frac{g'(t)}{1+g(t)}H(g(t))g'(t)dt.$$

$\square$

*Proof of Lemma 3.* To simplify notation, we will drop the index $i$ and treat $\theta$ as a scalar. The general proof for $\theta \in \mathbb{R}^c$ follows by repeating the proof for each $\theta_i$, while holding all other $\theta'_j s$ fixed.

$$\partial_\theta p_t(x;\theta) = \frac{1}{2}\mathbf{div}(p_t(x;\theta)\partial_\theta M_t(\theta)\nabla \log p_t(x;\theta))$$
$$= \frac{1}{2}p_t(x;\theta)(\mathbf{div}(\partial_\theta M_t(\theta)\nabla \log p_t(x;\theta)) + \langle \nabla \log p_t(x;\theta), \partial_\theta M_t(\theta)\nabla \log p_t(x;\theta)\rangle).$$

Dividing both sides by $p_t(x;\theta)$ gives

$$\partial_\theta \log p_t(x;\theta) = \frac{1}{2}(\mathbf{div}(\partial_\theta M_t(\theta)\nabla \log p_t(x;\theta)) + \langle \nabla \log p_t(x;\theta), \partial_\theta M_t(\theta)\nabla \log p_t(x;\theta)\rangle)$$
$$= \frac{1}{2}\sum_i \langle \partial_\theta M_t(\theta)e_i, \nabla^2 \log p_t(x;\theta)e_i\rangle + \frac{1}{2}\langle \nabla \log p_t(x;\theta), \partial_\theta M_t(\theta)\nabla \log p_t(x;\theta)\rangle$$
$$= \frac{1}{2}\sum_i \langle \partial_\theta M_t(\theta)e_i, \partial_c\nabla \log p_t(x + ce_i;\theta)\rangle + \frac{1}{2}\langle \nabla \log p_t(x;\theta), \partial_\theta M_t(\theta)\nabla \log p_t(x;\theta)\rangle.$$

Taking a derivative wrt $x$ gives

$$\partial_\theta \nabla \log p_t(x;\theta) = \frac{1}{2}\sum_i \langle \partial_\theta M_t(\theta)e_i, \partial_c\nabla^2 \log p_t(x + ce_i;\theta)\rangle + \langle \partial_\theta M_t(\theta)\nabla \log p_t(x;\theta), \nabla^2 \log p_t(x;\theta)\rangle$$
$$= \frac{1}{2}\sum_i \partial_r\partial_s\nabla \log p_t(x + re_i + s\partial_\theta M_t(\theta)e_i;\theta) + \partial_s\nabla \log p_t(x + s\partial_\theta M_t(\theta)\nabla \log p_t(x;\theta);\theta).$$

$\square$

**Lemma 5.**

$$\partial_\theta \texttt{flow}(x,t,\phi) = \frac{1}{2}\sum_{i=1}^d \partial_r\partial_s \texttt{flow}(x + re_i + s\partial_\theta M_t(\theta)e_i, t, \phi)$$
$$+ \partial_s \texttt{flow}(x + sM_\theta^{1/2}\partial_\theta M_t(\theta)\texttt{flow}(x,t,\phi), t, \phi) + \frac{1}{2}M_t^{-1}(\theta)(\partial_\theta M_t(\theta))\texttt{flow}(x,t,\phi).$$

*Proof of Lemma 5.* Recall that

$$\texttt{flow}(x;\theta) = M_\theta^{1/2}\texttt{net}(x;\theta) = M_\theta^{1/2}\nabla \log p_t(x;\theta).$$

We verify that

$$\partial_\theta \texttt{flow}(x;\theta) = M_\theta^{1/2}\partial_\theta\texttt{net}(x;\theta) + \frac{1}{2}M_\theta^{-1/2}(\partial_\theta M_\theta)\texttt{net}(x;\theta)$$

$$= \frac{1}{2}\sum_i M_\theta^{1/2}\partial_r\partial_s\texttt{net}(x + re_i + s\partial_\theta M_t(\theta)e_i;\theta) + M_\theta^{1/2}\partial_s\texttt{net}(x + s\partial_\theta M_t(\theta)\texttt{net}(x;\theta);\theta)$$

$$+ \frac{1}{2}M_\theta^{-1/2}(\partial_\theta M_\theta)\texttt{net}(x;\theta)$$

$$= \frac{1}{2}\sum_i \partial_r\partial_s\texttt{flow}(x + re_i + s\partial_\theta M_t(\theta)e_i;\theta) + \partial_s\texttt{flow}(x + sM_\theta^{-1/2}\partial_\theta M_t(\theta)\texttt{flow}(x;\theta);\theta)$$

$$+ \frac{1}{2}M_\theta^{-1}(\partial_\theta M_\theta)\texttt{flow}(x;\theta).$$

$\square$

**Lemma 6.** *Let $\tilde{u}$ be as defined in Section 5.1. Then*

$$\int_t^{t'} \tilde{u}(\bar{t};x,\hat{x},t,\hat{t})d\bar{t} = (M_{t_{k-1}}^{1/2} - M_{t_k}^{1/2})(\texttt{flow}(\tilde{x}_{t_k}, t_k))$$

$$- \frac{1}{2}(M_{t_{k-1}}^{1/2} - M_{t_k}^{1/2})^2(M_{\hat{t}_k}^{1/2} - M_{t_k}^{1/2})^{-1}\big(\texttt{flow}(\hat{x}_{\hat{t}_k}, \hat{t}_k) - \texttt{flow}(\tilde{x}_{t_k}, t_k)\big).$$

*Proof.* It suffices to verify that

$$\int_t^{t'} (\partial_t M_{\bar{t}}^{1/2})(M_{\bar{t}}^{1/2} - M_t^{1/2})d\bar{t}$$

$$= \int_t^{t'} \frac{1}{2}\partial_t M_{\bar{t}} - (\partial_t M_{\bar{t}}^{1/2})M_t^{1/2}dt$$

$$= \frac{1}{2}(M_{t'} - M_t) - \left(M_{t'}^{1/2}M_t^{1/2} - M_t^{1/2}M_t^{1/2}\right)$$

$$= \frac{1}{2}(M_{t'}^{1/2} - M_t^{1/2})^2.$$

This concludes the proof. $\square$

**Lemma 7.** *If $x_t$ evolves as the following (1. and 2. are equivalent):*

$$(1.)\ (x_t - x_0) \sim \mathcal{N}(0, M_t(\theta)), \qquad (2.)\ dx_t = -\frac{1}{2}\partial_t M_t(\theta)\nabla \log p_t(x_t;\theta)dt,$$

*where $p_t(x;\theta) := p_0 * \mathcal{N}(0, M_t(\theta))$, then the score above is the conditional expectation*

$$\nabla \log p_t(x;\theta) = M_t^{-1}(\theta)\mathbb{E}_{x_0|x_t=x}\left[x_0 - x_t\right],$$

*where $(x_0, x_t)$ are defined by the joint distribution $x_0 \sim p_0$ and $x_t = x_0 + \mathcal{N}(0, M_t)$.*

*Proof of Lemma 7.* The expression for $p_t$ is given by

$$p_t(x) = \int p_0(x_0)\frac{|M_t|^{-0.5}}{\sqrt{2\pi}}\exp\left(-0.5(x_0 - x)^\top M_t^{-1}(x_0 - x)\right)dx_0 = \int p_0(x_0)p_t(x|x_0)dx_0.$$

Taking the derivative of $\log p_t(x)$ with respect to $x$

$$\nabla_x \log p_t(x) = \frac{1}{p_t(x)}\nabla_x p_t(x)$$

$$= \frac{M_t^{-1}}{p_t(x)}\int p_0(x_0)\frac{|M_t|^{-0.5}}{\sqrt{2\pi}}\exp\left(-0.5(x_0 - x)^\top M_t^{-1}(x_0 - x)\right)(x_0 - x)dx_0$$

$$= M_t^{-1}\mathbb{E}_{x_0|x_t=x}\left[x_0 - x_t\right].$$

$\square$

## C   IMPLEMENTATION DETAILS OF $g_i$

For a fixed set of node locations $0 = \tau_0 < \tau_1 < \cdots < \tau_{K-1} = T$, the subspace noise schedule is defined in log–space between the smallest and largest variance values $g(0) = g_0$ and $g(T) = T$. The trainable parameters $\theta = (\theta_1, \ldots, \theta_{K-1})$ define a collection of strictly positive increments

$$s_i = \text{softplus}(\theta_i), \qquad i = 1, \ldots, K-1,$$

These increments are then rescaled by $\alpha$ so that their sum exactly matches the total log–gap between the endpoints,

$$\alpha = \frac{\log T - \log g_0}{\sum_{i=1}^{K-1} s_i}.$$

The log–values at the nodes are constructed by cumulative summation,

$$\ell_0 = \log g_0, \qquad \ell_j = \ell_0 + \sum_{i=1}^{j} \alpha\, s_i, \quad j = 1, \ldots, K-1,$$

so that $\ell_{K-1} = \log T$.

Given a time $t \in [0, T]$, one locates the enclosing interval $[\tau_{j-1}, \tau_j]$ and computes the normalized position

$$p(t) = \frac{t - \tau_{j-1}}{\tau_j - \tau_{j-1}}.$$

The value of $g(t)$ is then obtained by linearly interpolating between successive log–nodes and exponentiating:

$$\log g(t) = (1 - p(t))\, \ell_{j-1} + p(t)\, \ell_j, \qquad g(t) = \exp\big(\log g(t)\big).$$

Within each interval the derivative takes the simple form

$$g'(t) = g(t)\, \frac{\ell_j - \ell_{j-1}}{\tau_j - \tau_{j-1}}.$$

## D   REDUCING BIAS IN LEARNING $r(t; \gamma)$

Let $r$ be short for $r(t; \gamma)$. We defined in (19) that

$$\tilde{L}(\gamma) = \int_0^T \mathbb{E}_{x_0, \epsilon} \left[ \|\bar{v}(x_r, r; \theta, \phi) - \tilde{v}(x_r, x_0, r; \theta)\|_2^2 \right] dt$$

$$= \int_0^T \mathbb{E}_{x_0, \epsilon} \left[ \left\| (I + M_r)^{-1/2} \partial_t M_r \left( M_r^{-1}(x_0 - x_r) - \text{net}(x_r, t) \right) \right\|_2^2 \right]$$

$$= \int_0^T \mathbb{E}_{x_0, \epsilon} \left[ \underbrace{(x_0 - x_r)^\top M_r^{-1} A_r M_r^{-1}(x_0 - x)}_{\circledast} - 2(x_0 - x_r)^\top M_r^{-1} A_r \text{net}(x_r, t) + \text{net}(x_r, t)^\top A_r \text{net}(x_r, t) \right],$$

where we define $A_r := \partial_t M_r (I + M_r)^{-1} \partial_t M_r$. The purpose of $L$ is to capture the *score-matching loss*, as described in Section 3. However, recall that the *true* score-matching loss is really

$$\int_0^T \mathbb{E}_{x_r} \left[ \|v(x_r, r; \theta) - \bar{v}(x, t, \phi)\|_2^2 \right] dt$$

$$= \int_0^T \mathbb{E}_{x_0, \epsilon} \left[ \nabla \log p_r(x_r)^\top M_r^{-1} A_r M_r^{-1} \nabla \log p_r(x_r) - 2 \nabla \log p_r(x_r)^\top M_r^{-1} A_r \text{net}(x_r, t) + \text{net}(x_r, t)^\top A_r \text{net}(x_r, t) \right].$$

Contrasting the above with $\tilde{L}$, we see that $\tilde{L}$ additionally includes the variance of $M_{r(t;\gamma)}^{-1}(x_0 - x_{r(t;\gamma)})$ due to $\circledast$. For the purpose of optimizing $\gamma$, this additional variance introduces a non-trivial bias to the score-matching loss at time $t$.

To reduce this bias, we can instead approximate $\mathbb{E}_{x_0,\epsilon}\left[\nabla \log p_r(x_r)^\top M_r^{-1} A_r M_r^{-1} \nabla \log p_r(x_r)\right]$ by $\mathbb{E}_{x_0,\epsilon}\left[\texttt{net}(x_r,r)^\top M_r^{-1} A_r M_r^{-1} \texttt{net}(x_r,r)\right]$. This is based on the assumption that, even when $\texttt{net}$ is not a good approximation of $\nabla \log p$, $\|\texttt{net}\|_2$ should be a reasonably good approximation of $\|\nabla \log p\|_2$.

Consequently, we replace $\bar{L}(\gamma)$ by

$$\hat{L}(\gamma) := \int_0^T \mathbb{E}_{x_0,\epsilon}\left[-2(x_0 - x_r)^\top M_r^{-1} A_r \texttt{net}(x_r,t) + 2\texttt{net}(x_r,t)^\top A_r \texttt{net}(x_r,t)\right]. \quad (20)$$

Note that although the first term of $\hat{L}$ also involves $(x_0 - x_r)$, there is no bias in expectation.

## E BOUNDING THE ESTIMATOR VARIANCE

### E.1 THEORY

Let $\epsilon$ denote a random variable satisfying $\mathbb{E}[\epsilon] = 0$, $\mathbb{E}[\epsilon\epsilon^\top] = I_{d\times d}$. Let $v(x,t,\phi)$ denote the exact time derivative:

$$v(x,t,\phi) := \frac{1}{2}\sum_{i=1}^d \partial_r \partial_s \texttt{flow}(x + re_i + s\partial_\theta M_t(\theta)e_i, t, \phi)$$

$$+ \partial_s \texttt{flow}(x + s\partial_\theta M_t^{1/2}(\theta)\texttt{flow}(x,t,\phi), t, \phi) + \frac{1}{2}M_t^{-1}(\theta)(\partial_\theta M_t(\theta))\texttt{flow}(x,t,\phi),$$

where $\log(M_t(\theta))$ is matrix logarithm. Let $\hat{v}(x,t,\phi)$ denote the Hutchinson Estimator of $v$, defined as

$$\hat{v}(x,t,\phi) := \frac{d}{2}\partial_r \partial_s \texttt{flow}(x + r\epsilon + s\partial_\theta M_t(\theta)\epsilon, t, \phi)$$

$$+ \partial_s \texttt{flow}(x + s\partial_\theta M_t^{1/2}(\theta)\texttt{flow}(x,t,\phi), t, \phi) + \frac{1}{2}M_t^{-1}(\theta)(\partial_\theta M_t(\theta))\texttt{flow}(x,t,\phi),$$

where $\epsilon \sim \mathcal{N}(0, I)$.

**Lemma 8.** *Assume curvature bound* $\left\|\nabla^2 \texttt{flow}(x,t,\phi)\right\|_2 \le C$. *Then for all* $x,t,\phi$,

1. $\mathbb{E}[\hat{v}(x,t,\phi)] = v(x,t,\phi)$.

2. $\mathbb{E}[\|\hat{v}(x,t,\phi) - v(x,t,\phi)\|_2] \le 2Cd$

*Proof.* Recall that $\nabla^2 \texttt{flow}(x,t,\phi)$ is the second-derivative tensor. Then

$$\mathbb{E}[\partial_r \partial_s \texttt{flow}(x + r\epsilon + s\partial_\theta M_t(\theta)\epsilon, t, \phi)]$$

$$=\mathbb{E}\left[\epsilon^\top \nabla^2 \texttt{flow}(x,t,\phi)\epsilon\right]$$

$$=\sum_{i,j}\left[\nabla^2 \texttt{flow}(x,t,\phi)\right]_{ij}\mathbb{E}[\epsilon_i\epsilon_j]$$

$$=\sum_{i,j}\left[\nabla^2 \texttt{flow}(x,t,\phi)\right]_{ij}\mathbb{1}\{i = j\}$$

$$=\sum_i\left[\nabla^2 \texttt{flow}(x,t,\phi)\right]_{ii}$$

$$=\sum_i e_i^\top \nabla^2 \texttt{flow}(x,t,\phi)e_i$$

$$=\frac{1}{2}\sum_{i=1}^d \partial_r \partial_s \texttt{flow}(x + re_i + s\partial_\theta M_t(\theta)e_i, t, \phi).$$

This proves the first equality.

To prove the second inequality,

$$\mathbb{E}\left[\|\hat{v}(x,t,\phi) - v(x,t,\phi)\|_2\right]$$

$$\leq \mathbb{E}\left[\|\hat{v}(x,t,\phi)\|_2\right] + \mathbb{E}\left[\|\hat{v}(x,t,\phi) - v(x,t,\phi)\|_2\right]$$

$$\leq \mathbb{E}\left[\|\hat{v}(x,t,\phi)\|_2\right] + \sum_{i=1}^{d} \sup_{\|u\|_2 \leq 1} \left\|u^\top \nabla^2 \texttt{flow}(x,t,\phi)u\right\|_2$$

$$\leq \mathbb{E}\left[\|\hat{v}(x,t,\phi)\|_2\right] + Cd$$

$$= \mathbb{E}\left[\epsilon^\top \nabla^2 \texttt{flow}(x,t,\phi)\epsilon\right] + Cd$$

$$\leq \mathbb{E}\left[\|\epsilon\|_2^2\right] \sup_{\|u\|_2 \leq 1} \left\|u^\top \nabla^2 \texttt{flow}(x,t,\phi)u\right\|_2 + Cd$$

$$\leq 2Cd$$

where the first inequality is by triangle inequality, the second inequality is by $\|e_i\|_2 = 1$, the third inequality is by our assumption, the fifth line is by linearity, and the last line is by variance of standard Gaussian. This concludes the proof. $\square$

### E.2 EMPIRICAL VARIANCE BOUND

In the following, we evaluate the variance of estimating $\partial_\theta \texttt{flow}(x,t,\phi)$ using the Hutchinson Estimator. To be precise, let $s$ denote the dimension of $\theta$. In our experiment $g(t;\theta)$ is parameterized by its log-values at 32 nodes, so $s = 32$. Let $\nabla \in \mathbb{R}^s$ denote the true derivative of $L(\theta,\phi)$ wrt $\theta$, and let $\tilde{\nabla} \in \mathbb{R}^s$ denote the stochastic estimate using the Hutchinson Estimator. We define the *relative gradient error* as

$$\delta(x,t,\phi) = \|\frac{\nabla}{\|\nabla\|_2} - \frac{\tilde{\nabla}}{\|\tilde{\nabla}\|_2}\|_2.$$

With only $g_t(\theta)$, we can compute the ground truth $\nabla$ exactly by back-propagating through the network's time embedding. In the following table, we show the error $\delta(t) := \mathbb{E}_x\left[\delta(x,t,\phi)\right]$, where $t$ is fixed and $x$ is sampled randomly 50 times. For each sample, we draw $\epsilon \sim \mathcal{N}(0,I)$ for the Hutchinson estimator.

Table 2: Error in relative stochastic gradient estimate.

| $t$ | 6.4 | 1280 | 2560 | 3840 | 5120 | 6400 |
|---|---|---|---|---|---|---|
| $\delta(t)$ | 0.026 | $2.07e^{-7}$ | $3.84e^{-7}$ | $5.59e^{-7}$ | $3.76e^{-7}$ | $1.96e^{-7}$ |

## F ADDITIONAL EXPERIMENTS

### F.1 REVISIONS TO THE ORIGINAL EXPERIMENTS

For clarity, we updated the presentation of the small-NFE (Table 3) and large-NFE results (Figure 5) in the main paper. The three separate small-NFE tables for CIFAR-10, AFHQv2, and FFHQ have been merged into a single unified table (Table 1). In this unified table, we additionally report the minimum FID across three random seeds for each method. The appendix reports results from a single representative random seed.

All experimental settings for CIFAR-10 and FFHQ remain unchanged. The only minor adjustment is for AFHQv2, where we used a slightly larger regularization constant $c$ in the flow-matching loss to improve numerical stability. The motivation and analysis behind this choice are provided in Section F.3. This modification does not alter conclusions.

Overall trends across NFE and the relative performance of all methods remain consistent with the original submission. All ablation studies reported below were conducted using the original setting, and thus remain fully comparable to the originally reported results.

Table 3: FID ↓ vs. small NFE across datasets (50k samples).

| Method | CIFAR-10 | | | | | AFHQv2 | | | | | FFHQ | | | | |
|---|---|---|---|---|---|---|---|---|---|---|---|---|---|---|---|
| | 9 | 11 | 13 | 15 | 17 | 9 | 11 | 13 | 15 | 19 | 9 | 11 | 13 | 15 | 19 |
| EDM | 35.55 | 14.44 | 6.80 | 4.32 | 3.11 | 27.98 | 13.66 | 7.59 | 4.75 | 2.99 | 57.28 | 29.48 | 15.98 | 9.94 | 5.26 |
| $g^{iso}$ | 49.29 | 31.28 | 19.71 | 13.34 | 9.38 | 35.48 | 15.20 | 10.33 | 8.29 | 5.68 | 68.41 | 27.93 | 13.44 | 8.12 | 4.03 |
| $g_w^{iso}$ | 5.19 | 14.08 | 6.94 | **2.63** | 2.59 | **4.80** | 3.85 | 2.97 | 2.56 | 2.50 | 6.74 | 4.36 | 3.94 | 3.56 | 3.12 |
| $(g_1^{ani}, g_2^{ani})$ | 5.98 | **3.58** | **2.93** | 2.64 | **2.47** | 21.63 | 11.50 | 8.35 | 6.68 | 4.39 | 45.43 | 17.24 | 8.30 | 5.19 | 3.05 |
| $(g_{1,w}^{ani}, g_{2,w}^{ani})$ | **4.69** | 6.06 | 5.75 | 2.77 | 2.54 | 4.86 | **3.54** | **2.90** | **2.47** | **2.38** | **6.05** | **4.33** | **3.45** | **3.21** | **2.90** |

Table 4: FID ↓ of EDM and learned schedules across datasets.

| | **CIFAR-10** | | | | | | | | |
|---|---|---|---|---|---|---|---|---|---|
| Model | Schedule | nfe 9 | nfe 11 | nfe 13 | nfe 15 | nfe 17 | nfe 35 | nfe 59 | nfe 79 |
| **Ours** | $g_w^{iso}$ | 5.195 | 14.08 | 6.941 | 2.633 | 2.585 | **2.003** | 1.955 | 1.949 |
| **EDM** | $g_w^{iso}$ | 4.943 | 12.97 | 6.434 | **2.588** | 2.508 | 2.018 | **1.945** | **1.944** |
| **Ours** | $g_w^{ani}$ (geom.) | **4.687** | 6.060 | **5.753** | 2.769 | 2.536 | 2.078 | 2.082 | 2.039 |
| **EDM** | $g_w^{ani}$ (geom.) | 5.607 | **5.911** | 6.174 | 3.095 | 2.699 | 2.027 | 1.964 | 1.956 |

| | **AFHQv2** | | | | | | | | |
|---|---|---|---|---|---|---|---|---|---|
| Model | Schedule | nfe 9 | nfe 11 | nfe 13 | nfe 15 | nfe 19 | nfe 39 | nfe 79 | nfe 119 |
| **Ours** | $g_w^{iso}$ | 4.803 | 3.847 | 2.966 | 2.564 | 2.495 | 2.123 | 2.088 | 2.067 |
| **EDM** | $g_w^{iso}$ | 4.715 | 3.847 | 3.028 | 2.613 | 2.482 | 2.119 | 2.103 | 2.070 |
| **Ours** | $g_w^{ani}$ (geom.) | 4.859 | **3.542** | **2.897** | **2.472** | **2.376** | **2.087** | **2.055** | **2.035** |
| **EDM** | $g_w^{ani}$ (geom.) | **4.572** | 3.633 | 3.060 | 2.609 | 2.457 | 2.120 | 2.106 | 2.068 |

| | **FFHQ** | | | | | | | | |
|---|---|---|---|---|---|---|---|---|---|
| Model | Schedule | nfe 9 | nfe 11 | nfe 13 | nfe 15 | nfe 19 | nfe 39 | nfe 79 | nfe 119 |
| **Ours** | $g_w^{iso}$ | 6.737 | 4.365 | 3.935 | 3.558 | 3.122 | 2.418 | **2.335** | **2.346** |
| **EDM** | $g_w^{iso}$ | 7.182 | 4.771 | 4.042 | 3.704 | 3.397 | 2.629 | 2.498 | 2.482 |
| **Ours** | $g_w^{ani}$ (geom.) | **6.052** | **4.330** | **3.448** | **3.213** | **2.895** | **2.417** | 2.369 | 2.408 |
| **EDM** | $g_w^{ani}$ (geom.) | 6.723 | 4.945 | 3.864 | 3.555 | 3.225 | 3.370 | 3.405 | 3.653 |

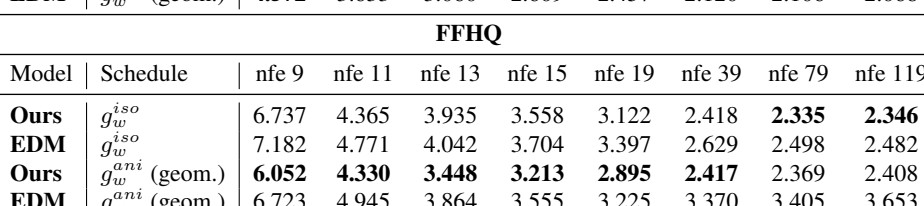
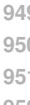
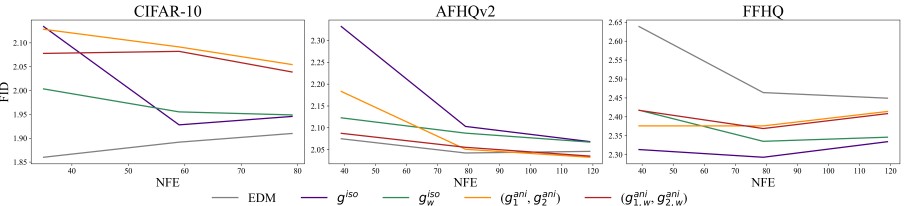

Figure 5: FID ↓ vs. large NFE across datasets (50k samples).

## F.2 EDM PERFORMANCE UNDER LEARNED SCHEDULES

We evaluate how the pretrained EDM model performs when used together with our learned schedules. Specifically, instead of sampling with the original EDM schedule, we replace it with either (1) our learned $g_{iso}$ wrapper or (2) the geometric-mean anisotropic wrapper based on $(g_{ani}, h_{ani})$. For each dataset, we report FID v.s. NFEs for: (i) the original EDM sampler, (ii) our model using the learned schedule, and (iii) the pretrained EDM model using the learned schedule.

As shown in Table 4, the pretrained EDM model exhibits a performance trend under our learned schedules that closely matches the trend observed when the same schedules are used with our model trained jointly with them. This consistency shows that the learned schedules not only benefit our own model but also improve the NFE performance of the pretrained EDM model, indicating that the schedules possess strong generalizability and transfer well across different network architectures.

Table 5: FID ↓ for different values of the flow-matching regularization constant $c$ on FFHQ and AFHQv2.

| | | | | | | | | | |
|---|---|---|---|---|---|---|---|---|---|
| **AFHQv2** | | | | | | | | | |
| $c$ | Schedule | nfe 9 | nfe 11 | nfe 13 | nfe 15 | nfe 19 | nfe 39 | nfe 79 | nfe 119 |
| **0.5** | $(g_1^{ani}, g_2^{ani})$ | 21.94 | 11.66 | 8.574 | 6.953 | 4.605 | 2.325 | 2.180 | 2.155 |
| **0.5** | $(g_{1,w}^{ani}, g_{2,w}^{ani})$ | 4.890 | 3.651 | 2.909 | 2.512 | 2.549 | 2.190 | 2.177 | 2.156 |
| **1** | $(g_1^{ani}, g_2^{ani})$ | 21.63 | 11.50 | 8.352 | 6.684 | 4.387 | 2.183 | 2.051 | 2.032 |
| **1** | $(g_{1,w}^{ani}, g_{2,w}^{ani})$ | 4.859 | **3.542** | 2.897 | 2.472 | **2.376** | 2.087 | 2.055 | 2.035 |
| **2** | $(g_1^{ani}, g_2^{ani})$ | 22.20 | 11.67 | 8.498 | 6.755 | 4.406 | 2.167 | 2.039 | 2.023 |
| **2** | $(g_{1,w}^{ani}, g_{2,w}^{ani})$ | **4.697** | 3.655 | **2.888** | **2.445** | 2.416 | **2.061** | **2.036** | **2.023** |
| **FFHQ** | | | | | | | | | |
| $c$ | Schedule | nfe 9 | nfe 11 | nfe 13 | nfe 15 | nfe 19 | nfe 39 | nfe 79 | nfe 119 |
| **0.5** | $(g_1^{ani}, g_2^{ani})$ | 43.12 | 16.37 | 7.963 | 5.046 | 3.040 | 2.388 | 2.381 | 2.423 |
| **0.5** | $(g_{1,w}^{ani}, g_{2,w}^{ani})$ | **5.862** | **4.130** | **3.392** | **3.183** | **2.833** | 2.422 | 2.370 | 2.412 |
| **1** | $(g_1^{ani}, g_2^{ani})$ | 45.43 | 17.24 | 8.297 | 5.192 | 3.052 | **2.376** | 2.376 | 2.414 |
| **1** | $(g_{1,w}^{ani}, g_{2,w}^{ani})$ | 6.052 | 4.330 | 3.448 | 3.213 | 2.895 | 2.417 | **2.369** | **2.408** |
| **2** | $(g_1^{ani}, g_2^{ani})$ | 42.51 | 16.06 | 7.788 | 4.964 | 3.023 | 2.400 | 2.389 | 2.426 |
| **2** | $(g_{1,w}^{ani}, g_{2,w}^{ani})$ | 6.015 | 4.324 | 3.494 | 3.241 | 2.860 | 2.450 | 2.385 | 2.421 |

## F.3 ABLATION ON THE REGULARIZATION CONSTANT $c$ OF FLOW-MATCHING LOSS

To understand the role of the regularization constant $c$ in the flow-matching objective, we evaluate training performance under three values, $c \in \{0.5, 1, 2\}$, on FFHQ and AFHQv2. For each setting, we report both the raw model performance $(g_1^{ani}, g_2^{ani})$ and the corresponding performance of $(g_{1,w}^{ani}, g_{2,w}^{ani})$ across NFEs (Table 5).

The results show a clear dataset-dependent pattern. For AFHQv2, $c = 2$ achieves the best performance consistently across NFEs for both the model and the wrapper. AFHQv2 exhibits substantially higher variation and more heterogeneous structures, so a stronger regularization term provides more stable gradients and improves training stability at low $t$. In contrast, FFHQ attains its best overall performance with $c = 1$, which offers a moderate level of stabilization while preserving flexibility in the learned flow. The $c = 0.5$ setting provides weaker regularization and underperforms on both datasets.

## F.4 ABLATION OF BASIS CHOICE

We evaluate the effect of the underlying basis used for anisotropic noise decomposition by training the wrapper $(g_{1,w}^{ani}, g_{2,w}^{ani})$ under three commonly used orthonormal bases: DCT, Haar wavelets, and PCA (computed from each dataset). All training and sampling settings are kept fixed to enable a controlled comparison.

Table 6 reports FID for CIFAR-10, AFHQv2, and FFHQ. For AFHQv2 and FFHQ, PCA performs slightly better at very small NFEs (e.g., 9–13), while DCT performs better as NFE increases. For CIFAR-10, the behavior is reversed: DCT performs best at small NFEs, whereas PCA becomes slightly better at larger NFEs. Haar underperforms across all settings. Although PCA can offer marginal improvements in certain low- or high-NFE regimes depending on the dataset, the DCT basis remains the most stable and reliable choice.

## F.5 ABLATION ON THE NUMBER OF IMAGE PASSES IN TRAINING FOR $(g_{1,w}^{ani}, g_{2,w}^{ani})$

We study how many image passes are needed to learn the anisotropic schedule $(g_{1,w}^{ani}, g_{2,w}^{ani})$ by training it under two settings: one with 100k/500k image passes per epoch and one with 2000k image passes per epoch, while keeping all other training configurations fixed. As shown in Table 7, the schedule obtained with the significantly smaller number of image passes (100k/500k) achieves

Table 6: FID $\downarrow$ of $(g_{1,w}^{ani}, g_{2,w}^{ani})$ using different bases (DCT, Haar, PCA) across datasets.

| | | **CIFAR-10** | | | | | | | |
|---|---|---|---|---|---|---|---|---|---|
| Basis | Schedule | nfe 9 | nfe 11 | nfe 13 | nfe 15 | nfe 17 | nfe 35 | nfe 59 | nfe 79 |
| **DCT** | $(g_{1,w}^{ani}, g_{2,w}^{ani})$ | **4.687** | **6.060** | **5.753** | **2.769** | **2.536** | **2.078** | 2.082 | 2.039 |
| **Haar** | $(g_{1,w}^{ani}, g_{2,w}^{ani})$ | 6.022 | 14.90 | 14.31 | 3.494 | 3.265 | 2.135 | 2.126 | 2.057 |
| **PCA** | $(g_{1,w}^{ani}, g_{2,w}^{ani})$ | 5.214 | 11.84 | 9.989 | 3.189 | 2.844 | 2.105 | **2.046** | **2.019** |

| | | **AFHQv2** | | | | | | | |
|---|---|---|---|---|---|---|---|---|---|
| Basis | Schedule | nfe 9 | nfe 11 | nfe 13 | nfe 15 | nfe 19 | nfe 39 | nfe 79 | nfe 119 |
| **DCT** | $(g_{1,w}^{ani}, g_{2,w}^{ani})$ | 4.859 | 3.542 | 2.897 | 2.472 | 2.376 | **2.087** | **2.055** | **2.035** |
| **Haar** | $(g_{1,w}^{ani}, g_{2,w}^{ani})$ | 7.276 | 3.744 | 3.026 | 2.478 | **2.299** | 2.286 | 2.234 | 2.267 |
| **PCA** | $(g_{1,w}^{ani}, g_{2,w}^{ani})$ | **4.598** | **3.101** | **2.588** | **2.383** | 2.330 | 2.117 | 2.070 | 2.079 |

| | | **FFHQ** | | | | | | | |
|---|---|---|---|---|---|---|---|---|---|
| Basis | Schedule | nfe 9 | nfe 11 | nfe 13 | nfe 15 | nfe 19 | nfe 39 | nfe 79 | nfe 119 |
| **DCT** | $(g_{1,w}^{ani}, g_{2,w}^{ani})$ | 6.052 | 4.330 | 3.448 | 3.213 | **2.895** | **2.417** | **2.369** | **2.408** |
| **Haar** | $(g_{1,w}^{ani}, g_{2,w}^{ani})$ | 8.558 | 6.539 | 4.717 | 3.337 | 4.490 | 2.898 | 2.391 | 2.447 |
| **PCA** | $(g_{1,w}^{ani}, g_{2,w}^{ani})$ | **5.448** | **3.818** | **3.382** | **3.162** | 3.016 | 2.647 | 2.656 | 2.733 |

Table 7: FID $\downarrow$ of $(g_{1,w}^{ani}, g_{2,w}^{ani})$ with different image passes across datasets.

| | | **CIFAR-10** | | | | | | | |
|---|---|---|---|---|---|---|---|---|---|
| kimgs | Schedule | nfe 9 | nfe 11 | nfe 13 | nfe 15 | nfe 17 | nfe 35 | nfe 59 | nfe 79 |
| **2000** | $(g_{1,w}^{ani}, g_{2,w}^{ani})$ | 4.687 | 6.060 | 5.753 | 2.769 | 2.536 | 2.078 | 2.082 | 2.039 |
| **500** | $(g_{1,w}^{ani}, g_{2,w}^{ani})$ | 4.778 | 6.951 | 5.641 | 2.956 | 2.542 | 2.124 | 2.078 | 2.050 |

| | | **AFHQv2** | | | | | | | |
|---|---|---|---|---|---|---|---|---|---|
| kimgs | Schedule | nfe 9 | nfe 11 | nfe 13 | nfe 15 | nfe 19 | nfe 39 | nfe 79 | nfe 119 |
| **2000** | $(g_{1,w}^{ani}, g_{2,w}^{ani})$ | 4.859 | 3.542 | 2.897 | 2.472 | 2.376 | 2.087 | 2.055 | 2.035 |
| **100** | $(g_{1,w}^{ani}, g_{2,w}^{ani})$ | 4.571 | 3.429 | 2.722 | 2.624 | 2.380 | 2.115 | 2.057 | 2.037 |

| | | **FFHQ** | | | | | | | |
|---|---|---|---|---|---|---|---|---|---|
| kimgs | Schedule | nfe 9 | nfe 11 | nfe 13 | nfe 15 | nfe 19 | nfe 39 | nfe 79 | nfe 119 |
| **2000** | $(g_{1,w}^{ani}, g_{2,w}^{ani})$ | 6.052 | 4.330 | 3.448 | 3.213 | 2.895 | 2.417 | 2.369 | 2.408 |
| **100** | $(g_{1,w}^{ani}, g_{2,w}^{ani})$ | 5.789 | 4.064 | 3.461 | 3.259 | 2.841 | 2.430 | 2.376 | 2.409 |

FID scores that closely match the results of the 2000k setting across all datasets and NFEs. This indicates that $(g_{1,w}^{ani}, g_{2,w}^{ani})$ can be learned very efficiently, requiring only a relatively small number of image passes to reach strong performance.

