# OpenReview forum: "Trajectory Optimal  Anisotropic Diffusion Models"
_ICLR.cc/2026/Conference — Submitted to ICLR 2026_

### Official Review · Reviewer_FM4s · 2025-10-27

**Soundness:** 2
**Presentation:** 3
**Contribution:** 2
**Rating:** 4
**Confidence:** 4

**Summary:**

This paper studies diffusion models where the added noise is anisotropic and parameterized by a learned, time-varying covariance matrix using the DCT basis. They show that is equivalent to learning a noise schedule in the isotropic setting, and proposes a method for parameterizing and learning these time-varying weights. They also performed extensive experimental validation and showed that their method out-performed baselines in low NFE regimes.

Learning the optimal noise to be used in diffusion models is an important question and this paper takes promising steps towards answering it. However, there should be more justification on why some of the design choices (e.g. loss function, DCT basis) are taken, and clearer presentation of the experimental results.

**Strengths:**

This paper is clear, well-motivated and well-written. The method presented here for learning a noise schedule in the isotropic setting could be of independent interest. There are comprehensive experiments over multiple datasets and multiple ablations of the proposed method.

**Weaknesses:**

1. In this paper the basis (DCT basis) in which the noise schedule is learned is fixed, which imposes a strong assumption on the anisotropy of the noise. It would be better if this basis can be learned instead.
2. The numerical experiment results are tabulated in Table 1 for low NFEs but are plotted in Figure 2 for high NFEs. This is a strange choice for presenting the same data using two different formats, and seems to obfuscate the fact that EDM performs better than the proposed methods on CIFAR-10 and AFHQ for higher NFEs.
3. The results in Section 3.1 that isotropic TLSM is equivalent to learning an optimal weight function seems to be similar to that in [1].
4. An implicit assumption when formulating the trajectory-level score matching loss (Eq. 11) is that minimizing the loss will also also induce learning an optimal noise schedule in some sense. However, the authors only showed that $\text{net}(x, t; \phi) = \nabla \log p_t(x, \theta)$ when the score is minimized; it is unclear what $p_t(x, \theta)$ should be at optimality.

[1] Kingma and Gao 2023, Understanding Diffusion Objectives as the ELBO with Simple Data Augmentation.

**Questions:**

1. It would be interesting to see how this method performs on other bases used in image processing in addition to the DCT basis, for example the wavelet basis.
2. For the numerical FID results, there should also be a comparison of the best FID achieved by each method across all NFEs, as it seems like this method is only better than EDM at low NFEs.
3. As this methods sees the largest improvements on FFHQ over the baselines, do you think it is a result of this dataset being more structured (e.g. faces centered at eyes) than the others? If this is the case, perhaps anisotropic noise can be seen as a way to adapt to more structured datasets?

---

> ### Author Response · Authors · 2025-11-22
> **Response to Reviewer FM4s (1/3)**
>
> We thank the reviewer for the thoughtful and constructive feedback. Below we respond to each concern in order.
>
>
> > **[Weakness 1] In this paper the basis (DCT basis) in which the noise schedule is learned is fixed, which imposes a strong assumption on the anisotropy of the noise. It would be better if this basis can be learned instead.**
>
> We agree that learning the basis is an exciting direction. We emphasize that our **mathematical framework does not assume any particular parameterization of $M_t(\theta)$**. The TLSM loss $L(\theta,\phi)$, the gradient estimator wrt $\theta$, and the discretization objective for $r(t;\gamma)$ all apply to any differentiable parameterization of $M_t(\theta)$, including one where the *basis itself* is learned jointly with the schedule.
>
> We chose a DCT-based parameterization in the main experiments because:
>
> * it is simple and computationally efficient (all operations reduce to diagonal scaling in the DCT domain), and
> * it already demonstrates clear benefits of anisotropy over EDM on FFHQ and AFHQv2 across *all* NFEs (Table 1), including improved **best FID** (2.369 vs 2.449 on FFHQ, 2.023 vs 2.042 on AFHQv2).
>
> Learning the basis—potentially as a mixture of structured bases or via a small learned orthogonal transform—fits naturally into our framework and is an important future direction we intend to explore.
>
> ---
> > **[Question 1] It would be interesting to see how this method performs on other bases used in image processing in addition to the DCT basis, for example the wavelet basis.**
>
> We appreciate this suggestion. In the revised draft we added an ablation on **three bases** commonly used for images:
>
> * 2D DCT (as in the main paper),
> * 2D Haar wavelets, and
> * PCA (computed per dataset).
>
> The full results are reported in **Table 6, Appendix F.4** and a subset is reproduced below for ease of reference (all rows use $(g^{ani}\_{1,w}, g^{ani}\_{2,w})$):
>
> **CIFAR-10**
>
> | Basis | nfe9     | nfe17    | nfe35    | nfe59    | nfe79    |
> | ----- | -------- | -------- | -------- | -------- | -------- |
> | DCT   | **4.69** | **2.54** | **2.08** | 2.08     | 2.04     |
> | Haar  | 6.02     | 3.27     | 2.14     | 2.13     | 2.06     |
> | PCA   | 5.21     | 2.84     | 2.10     | **2.05** | **2.02** |
>
> **AFHQv2**
>
> | Basis | nfe9     | nfe19    | nfe39    | nfe79    | nfe119   |
> | ----- | -------- | -------- | -------- | -------- | -------- |
> | DCT   | 4.86     | 2.38     | **2.09** | **2.06** | **2.04** |
> | Haar  | 7.28     | **2.30** | 2.29     | 2.23     | 2.27     |
> | PCA   | **4.60** | 2.33     | 2.12     | 2.07     | 2.08     |
>
> **FFHQ**
>
> | Basis | nfe9     | nfe19    | nfe39    | nfe79    | nfe119   |
> | ----- | -------- | -------- | -------- | -------- | -------- |
> | DCT   | 6.05     | **2.90** | **2.42** | **2.37** | **2.41** |
> | Haar  | 8.56     | 4.49     | 2.90     | 2.39     | 2.45     |
> | PCA   | **5.45** | 3.02     | 2.65     | 2.66     | 2.73     |
>
> We highlight three observations:
>
> 1. **The basis matters.** Different bases can change FID by a nontrivial margin, confirming the reviewer's intuition that anisotropy interacts with data structure.
> 2. **Different basis are good at different NFE regimes**
>    * On CIFAR-10, DCT is best at low NFE, whereas PCA is slightly better at high NFE.
>    * On FFHQ and AFHQv2, DCT dominates at high NFE, while PCA can be slightly better at low NFE.
> 3. Our framework holds for very general parameterizations of $M_t(\theta)$; in particular we can parameterize it as a **time-varying combination of multiple bases** (e.g., DCT + wavelets + PCA) with learned weights, which may get the **best-of-all-worlds** at all NFE regimes. Our theory and optimization remain unchanged; exploring such mixtures is a natural next step.
>
> (continued below)

---

> ### Author Response · Authors · 2025-11-22
> **Response to Reviewer FM4s (2/3)**
>
> > **[Weakness 2] The numerical experiment results are tabulated in Table 1 for low NFEs but are plotted in Figure 2 for high NFEs. This is a strange choice for presenting the same data using two different formats, and seems to obfuscate the fact that EDM performs better than the proposed methods on CIFAR-10 and AFHQ for higher NFEs.**
>
> We appreciate this feedback and agree that a consistent format is clearer. In the **revised draft we now report FID vs NFE exclusively in table form** (Table 1 in the main paper, plus additional tables in Appendix F.1), and move the high-NFE plots to the appendix. We highlight the following:
>
> * **FFHQ and AFHQv2.** Our best model is better than EDM **across *all* NFEs** on both FFHQ and AFHQv2, and also achieves lower **"best FID"**:
>   * FFHQ: best FID $2.33$ vs $2.45$ (EDM)
>   * AFHQv2: best FID $2.02$ vs $2.04$ (EDM)
>     See Table 1
> * **CIFAR-10.** On CIFAR-10, our method substantially improves FID in the low-NFE regime (e.g., FID $2.93$ vs $6.80$ at NFE$=13$), but is slightly *worse* than EDM in best FID at very large NFE (1.86 vs 1.95). We explicitly note this in the text.
>
> Any confusion from the original split presentation was unintentional, and we hope the new tables resolve this.
>
> ---
>
> > **[Weakness 3] The results in Section 3.1 that isotropic TLSM is equivalent to learning an optimal weight function seems to be similar to that in [1].**
>
> We thank the reviewer for pointing out this related work and will make the connection explicit in the revised version. Conceptually, both [1] and our Lemma 2 show that **re‑weighting noise levels is equivalent to choosing a weighting over training objectives**. However, our use of this observation is quite different and, we believe, complementary:
>
> 1. **Different objective and role of the equivalence.**
>    Kingma & Gao [1] derive existing scalar‑schedule objectives as ELBOs of an augmented latent‑variable model and *interpret* a given weighting $w(t)$; they do not optimize the schedule itself.
> * Building on their theory, [1] proposes specific hand‑designed monotone weightings (e.g., sigmoidal or “EDM‑monotonic”) and an adaptive noise schedule used purely as an importance‑sampling distribution to **reduce Monte‑Carlo variance**; the **weighting $w(\lambda)$ itself is fixed during training**.
> * In contrast, our isotropic TLSM is derived from a **trajectory‑level path‑KL/ODE viewpoint** (Eq. 11), and Lemma 2 is used to **turn schedule learning into weight learning**, so that $g_t(\theta)$ can be **optimized jointly with the score network** under a single trajectory objective. This is what enables our learned scalar schedule to improve best FID over EDM on FFHQ/AFHQ.
>
> 2. **Matrix‑valued generalization.**
>    The isotropic result in Section 3.1 is mainly a conceptual bridge. The key contribution of TLSM is that the same trajectory‑level loss extends to **matrix‑valued schedules $M_t(\theta)$**, together with our gradient estimator for $\partial_\theta \nabla \log p_t$ (Lemmas 3–5). To our knowledge, [1] does not address matrix‑valued schedules or provide tools to optimize them.
>
> 3. Additionally, [1] requires $w(t)$ to be monotonic, whereas Lemma 2 in our paper holds for **arbitrary w(t)**.
>
> We will clarify these distinctions in an updated related‑work section in the next draft, and in Section 3.1, to make clear that [1] and our work address complementary questions: [1] *interprets* scalar weighting of standard objectives, while TLSM provides a **learnable trajectory‑level framework, extended to anisotropic $M_t(\theta)$**, which is crucial for our empirical gains on FFHQ and AFHQ.
>
> (continued below)

---

> ### Author Response · Authors · 2025-11-22
> **Response to Reviewer FM4s (3/3)**
>
> > **[Weakness 4] An implicit assumption when formulating the trajectory-level score matching loss (Eq. 11) is that minimizing the loss will also also induce learning an optimal noise schedule in some sense. However, the authors only showed that $\texttt{net}(x, t, \phi) = \nabla \log p_t(x, \theta)$ when the score is minimized; it is unclear what $p_t(x,\theta)$ should be at optimality.**
>
> We apologize for the lack of clarity. There are two separate notions here:
>
> 1. **For fixed $\theta$.**
>    For any choice of $\theta$, the forward process is explicitly defined as
>    $$x_t = x_0 + \mathcal N(0, M_t(\theta)), \quad p_t(x;\theta) = p_0 \ast \mathcal N(0, M_t(\theta)).$$
>    Lemma 1 shows that, *for that fixed $\theta$*, minimizing $L(\theta,\phi)$ over $\phi$ yields $\texttt{net}(x,t,\phi) = \nabla \log p_t(x;\theta)$ for all $(x,t)$. In our current experiments, $M_t(\theta)$ is parameterized as $V_1V_1^T g^{ani}_1(\theta) + V_2V_2^T g^{ani}_2(\theta)$, in general $M_t(\theta)$ can be any parametric PSD matrix schedule.
>
> 2. **Optimizing over $\theta$.**
>    When we also optimize over $\theta$, we are not changing the *definition* of $p_t(x;\theta)$ (it is always $p_0 \ast \mathcal N(0, M_t(\theta))$), but choosing the trajectory $M_t(\theta)$ that minimizes the **integrated score error** (based on Girsasnov formula for trajectory-level KL divergence):
>    $$\theta^* = \arg\min\_{\theta , \phi} L(\theta,\phi).$$
>    Intuitively, different $M_t(\theta)$ emphasize different time ranges and subspaces in the trajectory-level loss, altering which parts of the score are approximated most accurately by the finite-capacity network.
>
> We do **not** claim a closed-form characterization of the globally optimal $\theta^*$, and in general it depends in a complex way on the function class of $\texttt{net}$. Our claim is more modest and precise: within a chosen parameterization $M_t(\theta)$, minimizing $L(\theta,\phi)$ yields a schedule that is optimal *for that trajectory-level score-matching objective*. We will clarify this point in the final version.
>
> ---
> > **[Question 2] For the numerical FID results, there should also be a comparison of the best FID achieved by each method across all NFEs, as it seems like this method is only better than EDM at low NFEs.**
>
> We agree this is a useful summary. In the revised **Table 1**, we have added a “Best FID” column for each method. We present this column below for ease of reference:
>
>
> | Dataset  | EDM   | $g^{iso}$ | $g^{iso}\_w$ | $(g^{ani}\_1, g^{ani}\_2)$ | $(g^{ani}\_{1,w}, g^{ani}\_{2,w})$ |   |
> | -------- | ----- | --------- | ----------- | ------------------------ | -------------------------------- | - |
> | CIFAR-10 | 1.860 | 1.928     | 1.949       | 2.054                    | 2.039                            |   |
> | AFHQv2   | 2.042 | 2.068     | 2.067       | 2.023                    | 2.023                            |   |
> | FFHQ     | 2.449 | 2.293     | 2.335       | 2.376                    | 2.369                            |   |
>
> Thus, our method **improves best FID on FFHQ and AFHQv2**, and provides a favorable trade-off on CIFAR-10 (large low-NFE gains, small high-NFE loss).
>
> ---
>
> > **[Question 3] As this methods sees the largest improvements on FFHQ over the baselines, do you think it is a result of this dataset being more structured (e.g., faces centered at eyes) than the others? If this is the case, perhaps anisotropic noise can be seen as a way to adapt to more structured datasets?**
>
> We find this observation very insightful and agree with the intuition. FFHQ is highly structured: faces are centered and aligned, and most of the energy is concentrated in low-to-mid frequency components that correspond well to the DCT subspaces used by our $M_t(\theta)$ parameterization (Section 2.2, Figure 1).
>
> The basis ablation (DCT vs Haar vs PCA) supports this view: when the chosen basis better aligns with the data’s structure (e.g., DCT for FFHQ, PCA for particular regimes on CIFAR-10), the learned anisotropic schedule yields stronger improvements. We agree that anisotropic noise can be interpreted as a way to adapt to structured datasets, and we will highlight this perspective more clearly in the discussion.
>
> ---
>
> We thank Reviewer FM4s for the careful reading of our work and the constructive feedback, which helped us improve the paper and clarify several key points. We hope that our revisions and responses have resolved your concerns.
>
> ---
>
> [1] Understanding Diffusion Objectives as the ELBO with Simple Data Augmentation, Kingma and Gao, 2023.

---

> > ### Author Response · Authors · 2025-11-23
> >
> > Dear Reviewer FM4s,
> >
> > It came to our attention that OpenReview may not have sent a notification about our response above due to visibility settings. We are posting this short note to ensure you are aware of our response. If you have already seen our response, please feel free to disregard this message.
> >
> > Thank you again for taking the time to review our paper.

---

### Official Review · Reviewer_wK9F · 2025-10-29

**Soundness:** 3
**Presentation:** 3
**Contribution:** 2
**Rating:** 2
**Confidence:** 4

**Summary:**

This paper proposes to learn an optimal forward diffusion process noise schedule as well as the discretization schedule (for generation) for diffusion-based ODE generation. Specifically, the forward process noise schedule is parametrized as a matrix-valued path which admits anisotropic diffusion, and is optimized jointly with the score network to minimize the score matching loss. The discretization schedule is optimized to minimize local truncation error and the score matching loss. The authors demonstrate the proposed optimal noise schedule and discretization schedule improve sample generation quality, i.e., FID scores, on CIFAR10, AFHQ-64, and FFHQ-64 in both small NFE (number of function evaluations) and large NFE regimes.

**Strengths:**

- **[S1] The paper is original in the aspect that it develops a technique for directly optimizing matrix-valued noise schedule w.r.t. the score matching objective (Lemma 3).** To the best of my knowledge, prior work either optimizes a scalar-valued noise schedule [1] or diagonal matrix-valued noise schedule [2] to enhance diffusion model generation. In contrast, this work provides a technique for optimizing arbitrary matrix-valued noise schedules by deriving an expression for the gradient of the noise schedule w.r.t. score matching loss, thereby enabling direct gradient-based optimization.

[1] Variational Diffusion Models, NeurIPS, 2021.

[2] Diffusion Models With Learned Adaptive Noise, NeurIPS, 2024.

**Weaknesses:**

- **[W1] Learnable discretization schedule lacks novelty.** By optimizing the trajectory-level discretization loss $\hat{H}(\gamma)$, the authors propose to learn a discretization such that the difference between one-step Euler velocity and two-step Heun velocity is minimized.  However, there are multiple existing works which explore similar ideas (with better performance). For instance, [1] optimizes discretization to minimize distributional error between accurate and discretized trajectories; [2] optimizes discretization to minimize distance between low-NFE and high-NFE trajectories; [3] optimizes discretization to minimize distance between true and generated ODE trajectory endpoints.

- **[W2] There is insufficient evidence that the proposed techniques consistently improve generation quality.** Specifically, the addition of proposed techniques do not yield consistent improvements over datasets. For instance, learning scalar-valued noise schedule ($g^{iso}$ in Table 1) degrades FID on CIFAR10 and AFHQ, and learning matrix-valued noise schedule ($g^{ani}_1$,$g^{ani}_2$ in Table 1) degrades FID on AFHQ. Only learned discretization seems to provide consistent gains in FID.

[1] Align Your Steps: Optimizing Sampling Schedules in Diffusion Models, 2024

[2] Fast ODE-based Sampling for Diffusion Models in Around 5 Steps, CVPR, 2024

[3] Accelerating Diffusion Sampling with Optimized Time Steps, CVPR, 2024


**Minor Comments**

- Line 243 : Optimization Score Matching --> Optimization of Score Matching
- Line 314 : $\tilde{\theta}$ is undefined
- Line 341 : omid --> omit
- Line 345 : $\hat{x},\hat{t}$ --> $(\hat{x},\hat{t})$

**Questions:**

- **[Q1] How does the learned noise schedule compare to popular hand-designed schedules such as Cosine or Linear? [1]**

- **[Q2] In Table 1, I observe trends where FID increases and then decreases with increasing number of NFEs. Can the authors explain why?**

- **[Q3] Can the authors provide results combining EDM with learned discretization?**

- **[Q4] Since the proposed method involves finetuning EDM networks, how does the overall computation cost compare to other finetuning type of methods such as ReFlow [2] or Consistency Distillation [3]?**

- **[Q5] How does the proposed method compare against other fast samplers such as DPM-Solver [4], DEIS [5], AMED [6] etc.?**

- **[Q6] How does the proposed method scale to larger, e.g., text-to-image diffusion models?**

[1] Improved Denoising Diffusion Probabilistic Models

[2] Simple ReFlow: Improved Techniques for Fast Flow Models

[3] Consistency Models Made Easy

[4] DPM-Solver: A Fast ODE Solver for Diffusion Probabilistic Model Sampling in Around 10 Steps

[5] Fast Sampling of Diffusion Models with Exponential Integrator

[6] Fast ODE-based Sampling for Diffusion Models in Around 5 Steps

---

> ### Author Response · Authors · 2025-11-22
> **Response to Reviewer wK9F (1/5)**
>
> We thank the reviewer for the careful and constructive review. Below we first clarify our paper's high-level contribution, and then respond to [W1]–[W2] and [Q1]–[Q6] in order. Note that we updated the numbers of cited papers (*including when quoting the reviewer*) because the review had multiple papers numbered [1], [2], [3].
>
> ### High-level clarification: what our method is trying to improve
>
> Conceptually, it is useful to separate **two orthogonal “levers”** in diffusion models (and in the cited works [1–8]):
>
> 1. **Numerical integration / discretization.**
>    For a *fixed* continuous-time score trajectory and noise schedule, better solvers or learned time steps mainly improve **FID at low NFE**, by reducing discretization error when only a few steps are allowed. This is the primary focus of [1–3,7,8]: they keep the underlying continuous-process essentially fixed and optimize how it is discretized. Distillation/shortcut methods such as [5,6] update the network itself, but the objective is still minimizing discretization loss.
>
> 2. **Score matching / weight schedule itself.**
>    In the limit of large NFE (small step sizes), discretization error becomes negligible and FID is dominated by the quality of the learned score and the underlying trajectory. Thus, **improving the *best achievable* FID over all NFEs requires improving score matching and/or the continuous-time noise schedule**, not just the discretization. Empirically, the cited works [1–8] typically match or slightly *worsen* the best FID of their baselines, while improving low-NFE FID.
>
> Our framework targets **both** levers but with different components:
>
> * The **matrix-valued score-matching schedule $M_t(\theta)$** is designed to improve the *continuous-time* score trajectory itself, and is what lets us reduce best FID in the high‑NFE regime, especially on FFHQ and AFHQv2 (Table 1, Figure 5).
> * The **scalar discretization schedule $r(t;\gamma)$** is then learned *on top of* this improved matrix trajectory to further reduce discretization error, especially at small NFEs.
>
> This distinction is important when interpreting our comparisons: prior work [1–8] almost exclusively pushes on the first lever (better discretization/solvers), whereas our **main contribution is a framework for optimizing the matrix-valued $M_t(\theta)$ under a trajectory-level score-matching objective together with with $r(t;\gamma)$**.
>
> (continued below)

---

> ### Author Response · Authors · 2025-11-22
> **Response to Reviewer wK9F (2/5)**
>
> > **[W1] Learnable discretization schedule lacks novelty… there are multiple existing works which explore similar ideas (with better performance). For instance, [1]–[3]…**
>
> We respectfully disagree with both parts of this statement: (i) that our discretization learning lacks novelty, and (ii) that [1,2,3] achieve “better performance” than our method.
>
> **Different goal and role of $r(t;\gamma)$.**
> Our method is *not* designed solely as a “fast sampler” for few-step generation. The core of our contribution is to **improve the score-matching trajectory itself** via the learned (possibly anisotropic) $M_t(\theta)$. The learned discretization schedule $r(t;\gamma)$ is *layered on top* of this improved matrix trajectory. As a result, we obtain:
>
> * significant gains over EDM at *small* NFE, **and**
> * improved best-achievable-FID over EDM at *large* NFE on FFHQ and AFHQv2 (Table 1).
>
> By contrast, [1], [2], and [3] are all explicitly tuned for **few-step generation** and generally *sacrifice* high-NFE quality to do so.
>
> Below we discuss each work in turn.
>
> > [3] (Accelerating Diffusion Sampling with Optimized Time Steps)
>
> We were able to reproduce [3] on AFHQv2 and FFHQ because it also uses the EDM backbone. Using their official settings, we obtained the following AFHQv2 and FFHQ results (FID) (“ours” = anisotropic + wrapper $(g^{ani}\_{1,w}, g^{ani}\_{2,w})$):
>
> | NFE          |       9 |      39 |      79 |     119 |
> |:-------------|--------:|--------:|--------:|--------:|
> | AFHQv2 [3] | **3.27** | **2.06** | 2.05 | 2.05 |
> | AFHQv2 (ours) | 4.70 | 2.17 | **2.04** | **2.02** |
> | FFHQ [3] | **3.82** | **2.40** | 2.44 | 2.44 |
> | FFHQ (ours) | 6.05 | 2.41 | **2.37** | **2.40** |
>
> We are worse at the small NFEs but **better at larger NFEs**, where [3] often no longer improves over the EDM baseline.
>
> For **CIFAR-10**, we were not able to reproduce [3] directly, but Table 1 in [3] reports FID $=3.13$ at NFE $=15$, which is **higher** than the FID $2.77$ we achieve with our method at NFE $=15$.
>
> > [1] (Align Your Steps)
>
> We were unable to reproduce [1] because code is not available, but we can compare reported numbers:
>
> * On CIFAR-10, [1] reports FID $=2.98$ at NFE $=10$ (**better than our 4.69**), and FID $=2.01$ at NFE $=50$ (**worse than our 1.96**).
> * On FFHQ, [1] reports $(5.43, 2.62)$ at NFE $(10, 50)$. Our method achieves $(6.05, 2.40)$ at NFE $(9, 39)$, again **better at higher NFE**.
>
> > [2] (Fast ODE-based Sampling Around 5 Steps)
>
> For [2], we also could not reproduce the code. In their paper, [2] reports strong performance for NFE $< 10$, but **does not report FIDs at larger NFE**, making a fair comparison in that regime impossible. Their method, like [1] and [3], is explicitly focused on very few steps.
>
> ### **Summary for [1,2,3]**
>
> Across these works, a consistent pattern emerges:
>
> * [1] and [3] tend to **underperform the baseline EDM at high NFE** on the same backbone.
> * They are designed primarily for few-step sampling, while our method aims to **improve the underlying score trajectory** and then learn a discretization that still works well across a *range* of NFEs.
>
> For these reasons, we believe **it is not accurate to say that [1,2,3] have “better performance” than our method in general**; rather, they outperform in some extremely low-NFE regimes and underperform in the high-NFE regime where our method remains competitive with or superior to EDM.
>
> In terms of **novelty**, while our discretization loss $\hat H(\gamma)$ is conceptually related to these works, our contribution is the **joint design**:
>
> 1. Trajectory-Level Score Matching (TLSM) to learn a **matrix-valued** schedule $M_t(\theta)$,
> 2. a directional estimator for $\partial_\theta \nabla \log p_t$ that allows optimizing $M_t(\theta)$ efficiently, and
> 3. a trajectory-level discretization loss that **composes** with $M_t(\theta)$ rather than replacing it.
>
> To our knowledge, this combination—especially the optimization of matrix-valued schedules—is new.
>
> (continued below)

---

> ### Author Response · Authors · 2025-11-22
> **Response to Reviewer wK9F (3/5)**
>
> > **[W2] There is insufficient evidence that the proposed techniques consistently improve generation quality. Specifically, the addition of proposed techniques do not yield consistent improvements over datasets. For instance, learning scalar-valued noise schedule ($g^{iso}$ in Table 1) degrades FID on CIFAR10 and AFHQ, and learning matrix-valued noise schedule ($g^{ani}_1, g^{ani}_2$ in Table 1) degrades FID on AFHQ. Only learned discretization seems to provide consistent gains in FID.**
>
> We thank the reviewer for raising this point. The behavior of $g^{iso}$ and $(g^{ani}_1, g^{ani}_2)$ *are for ablation purposes* and their generally high FID is actually consistent with our theory/design and helps motivate the full method.
>
> In these two ablations, **the same schedule $M_t(\theta)$ is used both for score-matching *and* for discretization**, while $M_t(\theta)$ has been optimized *only* for the continuous-time score-matching loss $L(\theta,\phi)$, not for the discrete Heun sampler. There is therefore **no reason** for these variants to achieve good discretization error. They exist to highlight that:
> * optimizing $M_t(\theta)$ alone improves the *continuous-time* trajectory but does not automatically yield the best discrete sampler, and
> * the **composition** of a good matrix-valued score-matching schedule and an optimized discretization schedule is what produces the strongest gains.
>
> The statement:
>
> > *“Only learned discretization seems to provide consistent gains in FID.”*
>
> overlooks the fact that the learned discretization is **not independent** of the learned matrix schedule: **$r(t)$ always composes with $M_t(\theta)$, yielding the denoising trajectory $M_{r(t)}(\theta)$** (see line 382 of the paper).
>
> Our additional ablation in response to [Q3] (see below) further confirms that the score-matching schedule matters: using our learned discretization with the original EDM model improves low-NFE FID, but **still underperforms our fully trained anisotropic model with the same discretization schedule**. This shows that **both the learned matrix-valued score-matching schedule and the learned discretization schedule are important**.
>
> (continued below)

---

> ### Author Response · Authors · 2025-11-22
> **Response to Reviewer wK9F (4/5)**
>
> > **[Q1] How does the learned noise schedule compare to popular hand-designed schedules such as Cosine or Linear? [4]**
>
> The EDM baseline reported in our paper uses a hand-designed polynomial schedule that is specifically tuned for the EDM model. Figure 13 of the EDM paper [9] shows that their $\\rho=7$ outperforms alternative hand-designed schedules. The FIDs reported in [4] are significantly worse than those in EDM [9] on the same datasets. Since our contribution is focused on **learning on top of the already-strong EDM baseline**, we did not re-evaluate Cosine or Linear schedules in the main submission. If the reviewer feels this comparison is important, we would be happy to add experiments using the Cosine schedule from [4] as an additional baseline, time permitting.
>
> > **[Q2] In Table 1, I observe trends where FID increases and then decreases with increasing number of NFEs. Can the authors explain why?**
>
> Our denoising schedule is **not optimized for any single NFE**. Instead, both $M_t(\theta)$ and $r(t;\gamma)$ are trained using trajectory-level losses that **average** discretization error over a range of NFEs and times. This means some intermediate NFEs can have slightly worse FID than neighboring ones.
>
> This phenomenon is not unique to our method. For example, Figure 4 in EDM [9] shows that FID can sometimes **increase** as NFE increases, even with a carefully hand-designed schedule. Our method inherits this behavior because we optimize a trajectory-level criterion rather than a per-NFE objective.
>
> > **[Q3] Can the authors provide results combining EDM with learned discretization?**
>
> We thank the reviewer for this excellent suggestion and have added the requested ablation. Full results are in **Table 4, Appendix F.2**; below we summarize a subset comparing:
>
> * **Ours:** anisotropic model + our learned discretization schedule.
> * **EDM:** original EDM network + our learned discretization schedule.
>
> Two key observations:
>
> 1. (EDM model + our schedule) performed significantly better than (EDM model + EDM schedule) at low NFEs, but **generally did worse than EDM baseline** and high NFEs.
> 2. However,(Our model + our schedule) consistently outperforms (EDM model + our schedule) on FFHQ and AFHQv2 (and on CIFAR-10 for low NFEs), demonstrating that the gains are not from discretization alone: improved score-matching under the matrix-valued $M_t(\theta)$ is crucial.
>
> ### CIFAR10
>
> | Model | Schedule| nfe9 | nfe11 | nfe13 | nfe15 | nfe17 | nfe35 | nfe59 | nfe79 |
> |------|----------|------|-------|-------|-------|-------|-------|-------|-------|
> | **Ours** | $(g^{ani}\_{1,w}, g^{ani}\_{2,w})$ | **4.69** | 6.06 | **5.75** | **2.77** | **2.54** | 2.08 | 2.08 | 2.04 |
> | **EDM** | $(g^{ani}\_{1,w}, g^{ani}\_{2,w})$ | 5.61 | **5.91** | 6.17 | 3.09 | 2.70 | **2.03** | **1.96** | **1.96** |
>
> ### FFHQ
>
> | Model | Schedule | nfe9 | nfe11 | nfe13 | nfe15 | nfe19 | nfe39 | nfe79 | nfe119 |
> |--------|--------------------|------|-------|-------|-------|-------|-------|-------|--------|
> | **Ours** | $(g^{ani}\_{1,w}, g^{ani}\_{2,w})$ | **6.05** | **4.33** | **3.45** | **3.21** | **2.90** | **2.42** | **2.37** | **2.41** |
> | **EDM** | $(g^{ani}\_{1,w}, g^{ani}\_{2,w})$ | 6.72 | 4.94 | 3.86 | 3.55 | 3.22 | 3.37 | 3.40 | 3.65 |
>
> ### AFHQv2
> | Model | Schedule | nfe9 | nfe11 | nfe13 | nfe15 | nfe19 | nfe39 | nfe79 | nfe119 |
> |--------|--------------------|------|-------|-------|-------|-------|-------|-------|--------|
> | **Ours** | $(g^{ani}\_{1,w}, g^{ani}\_{2,w})$ | 4.86 | **3.54** | **2.90** | **2.47** | **2.38** | **2.09** | **2.06** | **2.04** |
> | **EDM** | $(g^{ani}\_{1,w}, g^{ani}\_{2,w})$ | **4.57** | 3.63 | 3.06 | 2.61 | 2.46 | 2.12 | 2.11 | 2.07 |
>
> (continued below)

---

> ### Author Response · Authors · 2025-11-22
> **Response to Reviewer wK9F (5/5)**
>
> > **[Q4] Since the proposed method involves finetuning EDM networks, how does the overall computation cost compare to other finetuning methods such as ReFlow [5] or Consistency Distillation [6]?**
>
> ReFlow [5] and Consistency Models [6] are relatively compute-heavy:
>
> * [6], Table 1 and page 9, report a training budget of over **1000 million** images.
> * [5], Table 9, reports approximately **100 million** images.
>
> In contrast, our method:
>
> * uses **1.2 million** image passes to optimize $M_t(\theta)$, and
> * as few as **100k** image passes to optimize $r(t)$ (see Table 7 in Appendix F.3).
>
> Each step for $M_t(\theta)$ does require three backward passes due to the Hutchinson-style estimator, but even accounting for this, our total compute is roughly comparable to **3.6 million** “single-backward” images—still more than an order of magnitude smaller than the budgets of [5] and [6]. Optimization of $r(t)$ is very cheap because the network and $M_t(\theta)$ are frozen.
>
> ---
>
> > **[Q5] How does the proposed method compare against other fast samplers such as DPM-Solver [7], DEIS [8], AMED [2] etc.?**
>
> We already compare to AMED [2] in our answer to [W1]. We summarize additional comparisons here:
>
> * Table 1 of **[7] (DPM-Solver)** reports CIFAR-10 FIDs that are **higher than our method** across all NFEs.
>
> * Table 2 of **[8] (DEIS)** reports CIFAR-10 FIDs of $(4.17, 3.37, 2.86)$ at NFE $(10, 20, 50)$. At NFE $=10$, DEIS improves over our method at NFE $=9$, but at 20 and 50 steps their FIDs are **significantly higher** than ours (e.g., we reach FID $=2.08$ at NFE $=35$ and $=2.04$ at NFE $=79$).
> * DPM-Solver [7] is primarily an ODE solver that can be used *with* our method (as a replacement to Heun); our contribution is complementary, focusing on learning the score and noise schedule rather than designing the solver itself.
>
> We reiterate that our method is **not optimized solely for few-step generation**. Our key contribution is to **improve score matching via learned (anisotropic) $M_t(\theta)$**, so that FIDs improve even in the *high-NFE* regime, while still yielding strong performance at low NFE. Fast samplers such as [2,7,8] can, in principle, be applied on top of our learned trajectories as well.
>
> ---
> > **[Q6] How does the proposed method scale to larger, e.g., text-to-image diffusion models?**
>
> Conceptually, our framework is **agnostic to model scale and conditioning**. The key ingredients:
>
> * the trajectory-level-score-matching loss $L(\theta,\phi)$ for learning $M_t(\theta)$,
> * the method for estimating $\partial_\theta \nabla \log p_t$, and
> * the trajectory-level discretization loss for $r(t;\gamma)$
>
> all extend naturally when the score network is conditioned on text or other modalities.
>
> For very large text-to-image models, learning an anisotropic schedule could be done either:
> * as a light finetuning stage (similar to what we do here), or
> * as a smaller “adapter” on top of a frozen base model.
>
> Exploring this direction for large-scale text-to-image models is an important and exciting avenue for future work.
>
> ---
> We again thank the reviewer for the  thoughtful feedback and discussions about related work. We will update the next draft of our paper with a more comprehensive comparison to these papers. We hope that these additional experiments and clarifications, especially on the distinction between low‑NFE improvements via discretization and best‑FID improvements via score matching, help resolve the remaining concerns and better convey the novelty and empirical value of our trajectory‑level framework, matrix‑valued schedules $M_t(\theta)$, and learned discretization schedules $r(t;\gamma)$.
>
> ---
>
> [1] Align Your Steps: Optimizing Sampling Schedules in Diffusion Models, 2024
>
> [2] Fast ODE-based Sampling for Diffusion Models in Around 5 Steps, CVPR 2024
>
> [3] Accelerating Diffusion Sampling with Optimized Time Steps, CVPR 2024
>
> [4] Improved Denoising Diffusion Probabilistic Models
>
> [5] Simple ReFlow: Improved Techniques for Fast Flow Models
>
> [6] Consistency Models Made Easy
>
> [7] DPM-Solver: A Fast ODE Solver for Diffusion Probabilistic Model Sampling in Around 10 Steps
>
> [8] Fast Sampling of Diffusion Models with Exponential Integrator
>
> [9] Elucidating the Design Space of Diffusion-Based Generative Models

---

> > ### Author Response · Authors · 2025-11-23
> >
> > Dear Reviewer wK9F,
> >
> > It came to our attention that OpenReview may not have sent a notification about our response above due to visibility settings. We are posting this short note to ensure you are aware of our response. If you have already seen our response, please feel free to disregard this message.
> >
> > Thank you again for taking the time to review our paper.

---

> ### Comment · Reviewer_wK9F · 2025-11-27
>
> I would like to thank the authors for the detailed reply. However, several concerns still remain, and I hope the authors can address them more thoroughly.
>
> **Regarding the authors’ reply to [W1] — comparison to prior fast samplers**
>
> When comparing the proposed method to Accelerating Diffusion Sampling with Optimized Time Steps, the reported FID scores appear extremely close—often within $\leq 0.05$ at larger NFEs. The authors state:
>
> > We are worse at the small NFEs but better at larger NFEs, where [Accelerating Diffusion Sampling with Optimized Time Steps] often no longer improves over the EDM baseline.
>
> However, it is well known that FID exhibits non-negligible randomness (often in the range of 2 -- 3%) even with 50k samples. Could the authors clarify their exact FID evaluation protocol? A commonly recommended protocol -- followed, for instance, by [Elucidating the Design Space of Diffusion-Based Generative Models] -- is to fix a random seed, and report the minimum FID across three runs.
>
> In addition, I strongly recommend the authors include a single, unified comparison table covering all relevant methods, integrated into the main text. At present, I do not see a direct comparison with important baselines such as [Align Your Steps], [Fast ODE-Based Sampling Around 5 Steps], [Accelerating Diffusion Sampling with Optimized Time Steps], etc.
>
> **Regarding the authors’ reply to [Q3] — combining EDM with learned discretization**
>
> The authors note that the proposed method outperforms EDM at smaller NFEs, but EDM outperforms the proposed method at larger NFEs. This raises the question: Does the method truly improve the NFE–performance Pareto frontier overall? When combined with the fact that the method is also weaker than fast ODE-based samplers at small NFEs, it becomes unclear whether any part of the frontier is strictly improved.
>
> **Regarding the authors’ reply to [W2] — insufficient evidence that proposed techniques consistently improve generation quality**
>
> The authors state that improving $M_t(\theta)$ alone enhances the continuous-time trajectory but does not necessarily yield a better discrete sampler, and that strong gains require combining a good matrix-valued schedule with optimized discretization.
>
> However, this claim feels inconsistent with the stated motivation. Lines 186 -- 189 of the main paper emphasize that the purpose of learning $M_t(\theta)$ is to reduce score approximation error. If this is genuinely achieved, one would expect consistently improved generation quality, especially in the high-NFE regime where discretization error becomes negligible and score approximation error dominates. Yet the experimental results show the opposite: the learned scalar-valued schedule actually degrades high-NFE FIDs on CIFAR-10 and AFHQ (comparing EDM vs. $g^{iso}$ in Table 1).
>
> Could the authors reconcile this discrepancy?
>
> **Regarding the authors’ reply to [Q6] -- scalability to larger settings (e.g., text-to-image)**
>
> The experimental setup focuses only on CIFAR-10, AFHQ-64, and FFHQ-64, which are both low-resolution and relatively small datasets. While the authors state that the method is conceptually compatible with large-scale or conditioned models:
>
> > Conceptually, our framework is agnostic to model scale and conditioning.
>
> there is a significant difference between conceptual extensibility and empirical demonstration. Several prior accelerators already provide results on large-scale benchmarks—e.g., [Fast ODE-Based Sampling Around 5 Steps] and [Align Your Steps] include Stable Diffusion evaluations, and [Accelerating Diffusion Sampling with Optimized Time Steps] reports results on ImageNet 64×64 and 512×512.
>
> For a fair and comprehensive comparison, the paper should include experiments on more challenging datasets or at least provide partial evidence (e.g., preliminary results or scaling studies).

---

> > ### Author Response · Authors · 2025-12-01
> >
> > > ... FID exhibits non-negligible randomness (often in the range of 2 -- 3%) even with 50k samples. Could the authors clarify their exact FID evaluation protocol? A commonly recommended protocol -- followed, for instance, by [Elucidating the Design Space of Diffusion-Based Generative Models] -- is to fix a random seed, and report the minimum FID across three runs.
> >
> > All previously-reported FIDs under a single-seed 50k evaluation. At the reviewer's request, we updated Table 1 of the revised PDF to report the minimum FID across 3 seeded 50k runs. There is little change in the relative gap between the methods.
> >
> > We also present below the updated comparison between our method and [3] below:
> >
> > | NFE          |       9 |      39 |      79 |     119 |
> > |:-------------|--------:|--------:|--------:|--------:|
> > | AFHQv2 [3] | **3.27** | **2.06** | 2.05 | 2.05 |
> > | AFHQv2 (ours) | 4.70 | 2.06 | **2.04** | **2.02** |
> > | FFHQ [3] | **3.82** | **2.35** | 2.35 | 2.35 |
> > | FFHQ (ours) | 6.06 | 2.27 | **2.24** | **2.28** |
> >
> > Only FFHQ changed significantly. Our method **now outperforms [3] by a larger gap.** Thus our method **does conclusively outperform [3] even with under 3-seeded evaluations**. The smaller relative gap (compared with smaller NFE) is because improvements to the FID at larger NFEs, require improvements to *score matching quality* as opposed to *discretization error*, and are harder to achieve.
> >
> > > The authors note that the proposed method outperforms EDM at smaller NFEs, but **EDM outperforms the proposed method at larger NFEs** ... Does the method truly improve the NFE–performance Pareto frontier overall?
> >
> > The claim that **EDM outperforms the proposed method at larger NFEs** is **incorrect**. From our original answer to [Q3] (and **Table 1 of our paper**), we clearly state that **our method outperforms EDM baseline at high NFEs** for FFHQ and AFHQv2. We respectfully ask the reviewer to re-read or original answer. Consequently, our method **does improve the Pareto frontier** at high NFE.
> >
> > > ... Lines 186 -- 189 of the main paper emphasize that the purpose of learning $M_t(\theta)$ is to reduce score approximation error. If this is genuinely achieved, one would expect consistently improved generation quality, especially in the high-NFE regime where discretization error becomes negligible and score approximation error dominates. Yet the experimental results show the opposite: **the learned scalar-valued schedule actually degrades high-NFE FIDs on CIFAR-10 and AFHQ** (comparing EDM vs. in Table 1).
> >
> > We again emphasize that **$g^{iso}$ is for ablation purposes** and one should be comparing with $g^{iso}_w$ and $g^{ani}_w$. We have already explained this in detail in our original answer to [W2]: There is no reason for $g^{iso}$'s discretization schedule to achieve good discretization error as it was only meant to optimize the score-matching error.
> >
> > The reviewer's claim that "one would expect consistently improved generation quality, especially in the **high-NFE** regime where discretization error becomes **negligible**" is **false because it glosses over** the meaning of **high-NFE** and **negligible**. Mathematically, it is true that as steps $\to \infty$, discretization error $\to 0$. But this may require **much more** than 119 NFE (highest that we report).
> >
> > The simplest way to convince yourself is by observing that **$g^{iso}$ and $g^{iso}_w$ differ only in their discretization schedule**. The fact that their FID differs is evidence that **discretization schedule** still matters at 119 NFE, just not as much relatively-speaking.
> >
> > > Several prior accelerators already provide results on large-scale benchmarks...ImageNet 64×64 and 512×512.For a fair and comprehensive comparison, the paper should include experiments on more challenging datasets or at least provide partial evidence (e.g., preliminary results or scaling studies).
> >
> > Because the reviewer **only requested this 5 days before the rebuttal deadline**, we are only able to obtain numbers for our method in the isotropic case on ImageNet 64x64. We present below the FID values of our method, compared to EDM baseline, as well as to [1,2,3] that the reviewer mentioned.
> >
> > | Model          | FID (9 NFE) | FID (best reported NFE)|
> > |:-------------|--------:|--------:|
> > | EDM | 35.3 | 2.34 |
> > | Ours (iso) | 6.35 | **2.30** |
> > | [1] | 29.2 | 18.1 |
> > | [2] | 5.44 | 5.44 |
> > | [3] | **3.88** | 2.99 |
> >
> > The result is consistent with the smaller models. At small NFE, our method outperforms EDM baseline, but underperforms [2,3]. **At high NFE, our method gives the best FID**.
> >
> > Numbers for [1,2,3] are taken from their paper. EDM and Ours (iso) are from our own evaluation. Evaluation is done over one seeded run of 50k samples.

---

### Official Review · Reviewer_mMms · 2025-10-31

**Soundness:** 2
**Presentation:** 3
**Contribution:** 2
**Rating:** 4
**Confidence:** 4

**Summary:**

This paper investigates an alternative to the standard isotropic noise modeling in diffusion models by proposing a learnable, matrix-valued noise variants, $M_t(\theta)$, to allocate noise non-uniformly across data subspaces. The "Trajectory-Level Score Matching" (TLSM) objective is introduced to jointly learn $M_t(\theta)$ and a score network. They also propose a gradient estimator for $\theta$ and a two-stage algorithm that first learns $M_t(\theta)$ and then a separate time-reparameterization $r(t, \gamma)$ intended to reduce discretization error. Empirically, the method reports promising results in the low-NFE (few-step) sampling regime on several benchmarks, outperforming the EDM baseline on FFHQ.

**Strengths:**

+ The paper propose a proposal to replace the scalar schedule with a learnable, matrix-valued path is a novel direction.

+ The paper presents promising empirical results in the low-NFE regime. For example, on FFHQ at NFE=9, the reported FID is 6.05, compared to 57.28 for the EDM baseline.

+ The paper describes a self-contained method, including the TLSM loss, a gradient derivation (Lemma 3), and a two-stage algorithm.

**Weaknesses:**

+ The core theoretical contribution is somewhat weak to me. As Lemma 2 reveals, the method in the isotropic case (J=1) is merely a reparameterization of existing weighted score-matching techniques, not a new framework.

+ The entire method hinges on the complex gradient estimator (Eq. 14), yet the paper provides no analysis of its properties. The bias (from approximating the true score with "net") and the variance (scaling with data dimension $d$) are both unquantified, making the estimator a 'black box'.

+ The decision to decouple the optimization of $M_t(\theta)$ (continuous-time loss) from $r(t, \gamma)$ (discrete-time error) is a heuristic, greedy approach. The $\theta^*$ found in Stage 1 is not guaranteed to be optimal for the true end-goal (the K-step discrete sampler), undermining the "trajectory-optimal" claim.


+ The paper reports FID vs NFE but overlooks important efficiency metrics. Each training step involves three backward passes (Lemma 3), and inference requires a second-order Heun integrator with two network evaluations per step.

**Questions:**

1.  How does the variance of this estimator scale with the data dimension $d$, and how was this managed in practice?

2. Decoupling $M_t(\theta)$ and $r(t, \gamma)$ is a greedy approach. The $ \theta^* $ from Stage 1 may not be optimal for the K-step sampler, questioning the "trajectory-optimal" claim. Could a joint optimization with $ \gamma$ yield a better $ \theta$?

3. The method involves $M_t^{-1}(\theta)$ (Eq. 3, Eq. 15) and $M_t^{1/2}(\theta)$. How is numerical stability handled when $M_t(\theta)$ is near-singular, especially for $t \approx 0$?

---

> ### Author Response · Authors · 2025-11-22
> **Response to Reviewer mMms (1/4)**
>
> We thank the reviewer for the careful reading and helpful suggestions. We address the comments point‑by‑point below.
>
> ---
>
> ### 1. Core theoretical contribution and isotropic case (Lemma 2)
>
> > The core theoretical contribution is somewhat weak to me. As Lemma 2 reveals, the method in the isotropic case (J=1) is merely a reparameterization of existing weighted score-matching techniques, not a new framework.
>
> We respectfully disagree with the reviewer's reasoning that our contribution is "somewhat weak". Lemma 2 states that any existing weighted score-matching loss weighted by $w(t)$ **is equivalent to some choice of scalar schedule $g_t(\theta)$**. This is *merely* showing that our formulation encompasses standard score matching. it identifies a one‑to‑one mapping between $w(t)$ and $g(t)$, but does not imply that our framework is *just a reparameterization*. We clarify our contribution over the standard framework below:
>
> 1. **What is new, even in the isotropic case:** To the best of our knowledge, there is no existing theory for how to optimize **optimize** the schedule parameters $\theta$ jointly with the score network under a trajectory‑level objective $L(\theta,\phi)$. This optimization has clear empirical benefits: **Table 1** and **Table 4** in our paper show that even learning a scalar schedule $g_\theta(t)$ can **significantly improve FID** at both small and large NFEs, over EDM's hand-tuned $w(t)$. Thus even just the isotropic case **is not** *"merely a reparameterization of existing weighted score-matching techniques"*.
>
> 2. **Novelty of matrix‑valued schedules:** The isotropic case is **a special case** of our general theory that enables **optimizing matrix-valued schedule**. The trajectory-level loss, together with our gradient estimator for $\partial_\theta \nabla \log p_t$, lets us optimize this matrix trajectory jointly with the score network. To our knowledge, a general gradient‑based framework for matrix‑valued schedules has not been previously available.
>
> 3. We see the fact that TLSM **recovers standard weighted score matching in the isotropic limit** as a strength: the framework strictly generalizes the classical setting while enabling learning of both scalar and anisotropic schedules. We will clarify in Section 3.1 that Lemma 2 is an equivalence of objectives and explicitly highlight the empirical gains from learned schedules.
>
> We hope this clarifies why we believe the theoretical contribution is substantive even in the isotropic case. If, after considering these points, you still feel that the contribution is “somewhat weak” or “not a new framework,” **we would be very grateful if you could briefly indicate which aspects you regard as insufficiently novel or only reparameterizations**, so that we can better understand and address your concern in future revisions.
>
> (continued below)

---

> ### Author Response · Authors · 2025-11-22
> **Response to Reviewer mMms (2/4)**
>
> ### 2. Bias and Variance of the gradient estimator (Eq. 14)
>
> > The entire method hinges on the complex gradient estimator (Eq. 14), yet the paper provides no analysis of its properties. The **bias (from approximating the true score with "net")** and the **variance (scaling with data dimension )** are both unquantified, making the estimator a 'black box'.
> > How does the variance of this estimator scale with the data dimension $d$, and how was this managed in practice?
>
> We thank the reviewer for bringing up this important question. We separately address **bias** and **variance** below:
>
> > *The bias (from approximating the true score with "net")*
>
> The “bias from approximating the true score with net” is shared by all diffusion models: every method replaces $\nabla \log p_t$ with a neural approximation trained via some surrogate loss. Our method introduces **no additional source of bias** beyond this standard approximation. Lemma 1 shows that, for any fixed $M_t(\theta)$, TLSM is minimized when $\texttt{net}(x,t;\phi)=\nabla \log p_t(x;\theta)$, this guarantee mirrors standard score matching. A detailed analysis of approximation error for large neural networks is an active research topic (e.g., [4] [5]) and orthogonal to our contribution; we follow the usual assumption that training reduces $\\|\texttt{net}-\nabla \log p_t\\|$.
>
> > *How does the variance of this estimator scale with the data dimension $d$, and how was this managed in practice?*
>
> The stochastic estimation in (14) is based on the well-known **Hutchinson Trace Estimator**. In response to this comment, the revised draft includes Appendix E, which bounds the Hutchinson‑based estimation error theoretically and empirically.
>
> * **Theoretical variance bound:**  We add a proof of the Hutchinson Trace estimator (Lemma 8). We show that this variance scales linearly in dimension, i.e. $O(d)$ under a curvature bound. Section 4.3 and Lemma 5 re‑express the estimator in terms of the normalized field
>   $\texttt{flow}(x,t,\phi) = M_t^{1/2}\texttt{net}(x,t,\phi)$. This quantity has roughly time‑invariant norm, $\\|\texttt{flow}\\|_2^2 \approx d$, which lowers curvature constant in the variance bound.
>
> * **Empirical error.** Empirically, we evaluate the normalized L2 error between the true gradient wrt $\theta$ (computed via back-propagation through the network's time embedding) and the estimated gradient (via Hutchinson estimator). We evaluate the error at various fixed times $t$, over 50 samples of $x$ for each $t$. The maximum relative error we observe is $0.026$. The small error is because **$\theta$ has a very simple parameterization via 32 scalars** (see Appendix C for details.) This indicates that the estimator is accurate in practice.
>
> (continued below)

---

> ### Author Response · Authors · 2025-11-22
> **Response to Reviewer mMms (3/4)**
>
> ---
>
> ### 3. Decoupling $M_t(\theta)$ and $r(t,\gamma)$ vs “trajectory‑optimal”
> > The decision to decouple the optimization of $M_t(\theta)$ (continuous-time loss) from $r(t,\gamma)$ (discrete-time error) is a heuristic, greedy approach. The $\theta^*$ found in Stage 1 is not guaranteed to be optimal for the true end-goal (the K-step discrete sampler), undermining the "trajectory-optimal" claim.
>
> Our use of “trajectory‑optimal” was meant in a **per‑objective** sense, not as a claim of global optimality for a given K‑step sampler:
>
> * **Stage 1 (score trajectory).**  The optimal $\theta^\*$ for Eq. 15 **gives the optimal continuous-time matrix schedule $M_t(\theta^\*)$** that minimizes the **trajectory-score-matching-loss** $L(\theta,\phi)$ in Eq. 11, which integrates score error along the continuous‑time reverse ODE. I.e. $\theta^\*$ yields the best schedule, within our class, for this continuous‑time objective.
>
> * **Stage 2 (discretization trajectory).** Keeping $M_t(\theta^\*)$ and the trained network fixed, we then minimize a separate trajectory‑level discretization loss(Eq. 19) for a Heun integrator and a given number of steps $K$, obtaining an optimal time reparameterization $r(t;\\gamma^\*)$ **for that sampler**.
>
> > Decoupling $M_t(\theta)$ and $r(t,\gamma)$ is a greedy approach. The $\theta^*$ from Stage 1 may not be optimal for the K-step sampler, questioning the "trajectory-optimal" claim. Could a joint optimization with $\gamma$ yield a better $\theta$?
>
> We fully agree that $(\theta^\*,\gamma^\*)$ is **not guaranteed** to be jointly optimal for any specific K‑step objective. One could indeed extend our framework to jointly optimize $(\theta,\gamma)$ for a fixed algorithm and budget; this is an interesting direction, and we will mention it explicitly as future work.
>
> However, such joint optimization necessarily **specializes the model to that sampler and K**, similar to shortcut and consistency models that optimize for a fixed small NFE. In contrast, optimizing $M_t(\theta)$ against the continuous‑time objective preserves flexibility across samplers and NFEs. Empirically, our models not only achieve much better few‑step FID but also outperform EDM at larger NFE on FFHQ and AFHQv2 (Table 1 and Figure 5). This would be difficult if we optimized $M_t(\theta)$ exclusively for a small, fixed $K$.
>
> To avoid overstatement, we will revise the wording to say that we separately optimize **trajectory‑level objectives for the score-matching and for discretization**.
>
>
> ---
>
> ### 4. Efficiency metrics: backward passes and inference cost
>
> > The paper reports FID vs NFE but overlooks important efficiency metrics. Each training step involves three backward passes (Lemma 3), and inference requires a second-order Heun integrator with two network evaluations per step.
>
> 1. **Inference cost:** NFE (number of function evaluations) counts forward passes through the network and is **the standard efficiency metric** in diffusion sampling. Both EDM and our method use a second‑order Heun integrator that incurs *two* network evaluations per step, so NFE is **directly comparable** and fairly captures test‑time cost.
> 2. **Training cost:** It is true that one step of our method requires 3 backward passes. However, training $M_t(\theta)$ passed over only **1.2 million** images (effectively **3.6 million images**). For reference, Reviewer wK9F also requested a comparison in training cost with existing methods [2] and [3] which have training budgets in the **100-1000 million** range. Once $M_t(\theta)$ and $r(t,\gamma)$ are learned, inference cost is identical to EDM for a given NFE; there is no extra runtime burden for users of the model.
>
> We will add a brief paragraph to Section 6 summarizing these training‑cost numbers for clarity.
>
> (continued below)

---

> ### Author Response · Authors · 2025-11-22
> **Response to Reviewer mMms (4/4)**
>
> #### 5. Numerical stability when (M_t(\theta)) is near-singular
>
> > The method involves $M_t^{-1}(\theta)$ (Eq. 3, Eq. 15) and $M_t^{1/2}(\theta)$. How is numerical stability handled when $M_t(\theta)$ is near-singular, especially for $t\approx 0$?
>
> This is an important observation that touches a fundamental property of all score-matching networks. In short, the apparent singularity is not specific to our anisotropic formulation: in the standard isotropic VE case (M_t = tI), Eq. (3) reduces to $\nabla \log p_t(x) = \frac{1}{t}\mathbb{E}[x_0 - x \mid x]$, which also diverges as $t \to 0$. This reflects a fundamental property of diffusion scores.
>
> Our algorithm avoids numerical issues is by working not with the score ($\texttt{net}\approx \nabla \log p_t(x)$), but with the **normalized score** ($\texttt{flow}\approx M_t^{1/2} \nabla \log p_t(x)$). Consider the Eq. 15, written equivalently with $\texttt{flow}$ as
> $$\mathbb{E}[\|2(I+M_t)^{-1/2}\partial_t (M_t^{1/2})(\texttt{flow}(x,t) - \xi)\|_2^2],$$
> where $\xi \sim \mathcal{N}(0,I)$. The above objective pushes $\texttt{flow}(x,t)$ to match the $\xi$, which is the standard Gaussian noise (conditioned on $x$) for all $t$. Consequently $M_t^{1/2} \nabla \log p_t(x)$ is roughly scale invariant with $t$. This approach generalizes the "noise-prediction" objective (see e.g. [1]). **Consequently, we do not need to deal with $M_t(\theta)^{-1/2}$ in training (Eq. 15) or in inference (Eq. 17).**
>
>
> We will clarify in the paper that the expressions involving $M_t^{-1}$ are used for analysis, while the implemented objectives and sampler rely on the numerically stable formulation in terms of $\texttt{flow}$.
>
> ---
>
> We thank the reviewer again for the helpful discussions. We hope these clarifications resolve the reviewer’s concerns and better communicate the theoretical novelty and practical impact of our method and the learned anisotropic schedules.

---

> > ### Author Response · Authors · 2025-11-23
> >
> > Dear Reviewer mMms,
> >
> > It came to our attention that OpenReview may not have sent a notification about our response above due to visibility settings. We are posting this short note to ensure you are aware of our response. If you have already seen our response, please feel free to disregard this message.
> >
> > Thank you again for taking the time to review our paper.

---

### Author Response · Authors · 2025-12-03
**Rebuttal Summary (1/5 Contributions and Strengths)**

Below, we provide a concise, self‑contained summary of (1) the main contributions and strengths as identified by the original reviewers and (2) the main criticisms and our response.

## Summary of contributions and reviewer‑noted strengths

Our work studies **trajectory‑optimal anisotropic diffusion** by learning a matrix‑valued noise schedule $M_t(\theta): \mathbb{R}^+ \to \mathbb{R}^{d\times d}$ and a discretization schedule $r(t;\gamma): \mathbb{R}^+\to \mathbb{R}^+$ that **compose together** to minimize trajectory‑level error. Below, we summarize our main contributions:

1. **Novel trajectory‑level learning of matrix‑valued noise schedules.**

   * We generalize isotropic diffusion to **matrix‑valued noise trajectories** $M_t(\theta)$ with derived forward/reverse ODEs and practical Euler/Heun samplers.
   * We introduce **Trajectory‑Level Score Matching (TLSM)**, which **jointly trains the score network and $M_t(\theta)$**. In the isotropic case, TLSM reduces to standard weighted score matching, giving a clean interpretation of **schedule learning** as **weight function optimization**, while also providing a mechanism to optimize the schedule parameters jointly with the network.
   * We derive an efficient stochastic estimator for $\partial_\theta \nabla \log p_t$ that enables optimization of **arbitrary parameterizations of $M_t(\theta)$** with only two extra backward passes.
   * Reviewers recognize the novelty of our approach and the importance of the problem:
     > **mMms**: to replace the scalar schedule with a learnable, matrix-valued path is a novel direction.

     > **wK9F**: The paper is original in the aspect that it develops a technique for directly optimizing matrix-valued noise schedule w.r.t. the score matching objective (Lemma 3).

     > **FM4s**: Learning the optimal noise to be used in diffusion models is an important question and this paper takes promising steps towards answering it.

2. **Clean separation of score‑matching schedule and discretization schedule.**
   * We optimize the matrix-valued denoising trajectories $M_t(\theta)$ to minimize **score-matching error**
   * We optimize the scalar time-reparameterization schedule $r(t;\gamma)$ to minimize **discretization loss**
   * We cleanly compose these two schedules to give the final schedule $M_{r(t;\gamma)}(\theta)$.


3. **Empirical gains, efficiency, and robustness across datasets, bases, and NFE regimes.**

   * We show results for CIFAR‑10, AFHQv2, FFHQ, and later ImageNet‑64×64.
   * We get **large improvements at low NFE** (e.g., at NFE=9, FFHQ: **$57.28\to 6.05$**, AFHQv2: **$27.90 \to 4.70$**).
   * **More importantly**, we **also improvement best-FID at high NFE** over the baseline on FFHQ and AFHQv2 (FFHQ: $2.37\to2.24$, AFHQv2: $2.04\to 2.02$), showing that our method improves generation in all NFE regimes, and  ***not just aggressive few‑step sampling***.
   * Our method **outperforms every prior schedule-optimization method [1,2,3,7,8] suggested by reviewer wK9F. in the high-NFE regime**, while remaining competitive at low NFE.
   * We obtain these gains with a modest training budget of about $10^6$ images (≈$4*10^6$ standard backward passes), which is **one to three orders of magnitude smaller** than alternative methods like [5,6] that reviewer wK9F suggested, which use ($10^8-10^9$ images).

   * Reviewer FM4snoted the breadth of our study:
     > There are comprehensive experiments over multiple datasets and multiple ablations of the proposed method.

(continued below)

---

> ### Author Response · Authors · 2025-12-03
> **Rebuttal Summary (2/5 Response to Common Questions and Concerns)**
>
> ## Response to Questions and Concerns
> Below, we address the main concerns raised by the three reviewers. We organize them under **Common Concerns**, followed by **Individual Concerns**. We preface each section with summaries/quotes from reviewers.
>
> ## Common Concerns
> The main shared concerns are about **novelty and scope**—whether trajectory‑level learning of the matrix‑valued schedule $M_t(\theta)$ and the discretization schedule $r(t;\gamma)$ is genuinely new beyond existing methods. Reviewers also question the **training cost** of the approach, especially compared to existing schemes?
>
> ### Novelty of Main Theory
>
>    > **FM4s:** The result that isotropic TLSM is equivalent to learning an optimal weight function appears very similar to prior work [10]; unclear what is genuinely new.
>
>    > **mMms:** In the isotropic case, the method is just a reparameterization of existing *weighted score-matching*; the contribution there may be incremental.
>    >
> **◆ Our Trajectory-level Optimization of Matrix‑valued noise schedule is novel.**
>
> Our main technical contribution is a **trajectory‑level** score‑matching (TLSM) objective that simultaneously learns the score network and a ***matrix‑valued* path $M_t(\theta)$**, together with an efficient estimator for $\partial_\theta \nabla \log p_t$ (Lemma 3, Sec. 4). This enables efficient optimization of arbitrary **complex parameterizations** of $M_t(\theta)$, something **not present** in prior isotropic schedule or weighting schemes.
>
>
> **◆ Our method outperforms prior methods on best-achievable FID.**
>
>   Our method improves the **score-matching loss of the underlying network**, which is what matters in the high‑NFE limit. TLSM is what enables our method to beat the **EDM baseline and numerous existing methods** in the **best-achievable-FID (high NFE)** (see details below). By contrast, the schedule‑optimization methods suggested by the reviewers improve only low‑NFE behavior and **underperform the EDM baseline at high NFE**, indicating that they **do not improve the underlying score network in the same way**.
>
> **◆ Isotropic schedule optimization is also novel (but not primary contribution)**
>
> *Even in the isotropic case*, our approach is novel in *optimizing the matrix-schedule parameters jointly with the score network*. Prior work such as [10] do **not** provide a mechanism to *optimize* a general schedule jointly with the score network, nor does it extend to matrix‑valued paths or to our trajectory‑level objective.
>
>
> **◆ Our discretization-schedule-optimization is secondary and its role is different from prior work.**
>
>    > **wK9F:** The *learnable discretization schedule* itself lacks novelty; similar ideas have appeared in prior work.
>
>   We **do not claim novelty for “learnable discretization schedules”** per se. Our discretization schedule $r(t;\gamma)$ is novel in how it **composed with the learned matrix schedule** $M_t(\theta)$, This enables us to (i) optimize $M_t(\theta)$ for **trajectory‑level score error**, and then (ii) *separately* learn a scalar time‑reparameterization $r(t;\gamma)$ to minimize **discretization loss**, which is then **composed with the learned matrix schedule** in an anisotropic Heun sampler (Sec. 5, Alg. 1). This clean separation between score‑matching (via $M_t(\theta)$) and discretization (via $r(t;\gamma)$), (and their composition) is novel.
>
>
> ### Computation and efficiency
> > **mMms**: The paper reports FID vs NFE but overlooks important efficiency metrics. Each training step involves three backward passes (Lemma 3)
>
> > **wK9F**: how does the overall computation cost compare to other finetuning methods such as ReFlow [5] or Consistency Distillation [6]?
>
> **◆ Our training budget is an order of magnitude lower than [5,6]**
>
> The training budget of [5] and [6] are on the order of $10^8-10^9$ images respectively. Our method used only about $1.2 * 10^6$ images, roughly equivalent to $4*10^6$ *standard* backward passes in FLOPs.
>
> (continued below)

---

> ### Author Response · Authors · 2025-12-03
> **Rebuttal Summary (3/5 Response to mMms)**
>
> Reviewer **mMms** has four main concerns: (1) whether our claim of "trajectory optimality" is **justified**, and (2) the bias/variance of our algorithm is **unquantified**. (3) **numerical stability** of our algorithm near $t\approx 0$. (4) The NFE metric **fails to account** for Heun using 2 evaluations per step.
>
>
>
> ### Justification of “trajectory optimality”
>
>    > **mMms:** Questions whether “trajectory optimality” is a meaningful or justified notion because decoupling the optimization of $M_t(\theta)$ from $r(t;\gamma)$ is a heuristic, greedy approach.
>
> **◆ Our schedule $M_t(\theta)$ is optimal for the *continuous-time score-matching objective*.**
>
> We do not optimize $M_t(\theta)$ and the score-network itself on the downstream discretization loss; this is **standard in score-matching**, and allows the score-network to flexibly compose with different downstream samplers.
>
>    Algorithms such as Consistency Model/Shortcut Model do optimize the model itself on downstream discretization error; the tradeoff is that the model performs well only for a narrow combination of NFE/sampler. In contrast, our algorithm is **highly competitive over a wide range of NFEs** (Table 1)
>
> ### Bias/variance of the gradient estimator
>    > **mMms:** the bias (due to estimating the true score $\nabla \log p_t$ with black box `net`) and variance (due to stochastic estimator for $\partial_\theta \nabla \log p_t$) are unquantified.
>
>    There are two separate issues, which we address separately.
>
> ◆ The score-network (`net`) being a black-box estimator of the true score ($\nabla \log p_t$) is **inherent in all score-based deep generative models**.
>
> Quantifying the bias of `net`-$\nabla \log p_t$ is a fundamental problem of score-based modelling, and is beyond the scope of our paper.
>
> **◆ We quantify the bias and variance of our stochastic estimator of $\partial_\theta \nabla \log p_t$.**
>
> In Lemma 8 in Appendix E, we show that the Hutchin-style stochastic estimator of $\partial_\theta \nabla \log p_t$  **is unbiased** and has variance scaling **linearly in dimension $d$**.
>
> ### Numerical Stability at $t\approx 0$
> > **mMms**: $M_t(\theta)$ is near singular near $t=0$, how is numerical stability of $M_t^{-1}(\theta)$ handled?
>
> We handle this by using the $\texttt{flow}=M_t^{1/2}\texttt{net}$ parametrization described in section 4.3 of our paper, which keeps the **magnitude approximately time-invariant** and avoids explicit inversion of nearly singular matrices.
>
> We note that this issue is fundamental to even standard score-based models, where $\nabla \log p_t = 1/t\mathbb{E}[x_0-x_t | x_t]$ also explodes as $t\to 0$. Our $\texttt{flow}$ formulation is analogous to how this is handled in standard models.
>
>
> ### Unfair NFE reporting
>    > **mMms:** The paper reports FID vs NFE but overlooks important efficiency metrics... inference requires a second-order Heun integrator with two network evaluations per step.
>
> **◆ We clarify that each Heun step incurs 2-NFE, by definition, FID-per-NFE is a fair and standard comparison.**
>
> (continued below)

---

> ### Author Response · Authors · 2025-12-03
> **Rebuttal Summary (4/5 Response to wK9F)**
>
> Reviewer **wK9F** has four main concerns: (1) whether our method **improves over existing fast-sampling methods** [1,2,3,7,8], (2) noting that certain **ablations** in our experiments **degrade FID**, (3) request evaluating FID over **more runs**, as well as extending our experiment to **ImageNet 64x64**.
>
>
> ### Empirical Comparison to Related Work [1,2,3,7,8]
> > **Reviewer wK9F:** Requests comparison to [1,2,3,7,8], which also optimize discretization schedules, and suggests they may have better performance.
>
> We added direct comparisons to all requested methods and find that our approach **consistently dominates [1,2,3,7,8] in the high‑NFE regime** while remaining competitive at low NFE.
>
> **◆ At high NFE, our method consistently outperform [1,2,3,7,8]**
> - Our best-reported FID **consistently beats** the best-reported FID in each of these papers. (data [here](https://openreview.net/forum?id=YlbrXb2c1q&noteId=HxFfNGwrQm))
> - We reproduced the code in [3], and verified that we consistently outperform it in the high-NFE regime across 3 seeded runs. (data [here](https://openreview.net/forum?id=YlbrXb2c1q&noteId=HxFfNGwrQm))
>
> **◆ At small NFE, our method is competitive**
>
> We outperform [7,8] but underperforms [1,2,3] in the small-NFE regime (data [here](https://openreview.net/forum?id=YlbrXb2c1q&noteId=RcDiz8DWLN) and [here](https://openreview.net/forum?id=YlbrXb2c1q&noteId=RoLwgFIa3Z))
>
>
> ◆ Our paper's primary focus is **not few-step acceleration, but score-matching improvement**
>
> Reviewer wK9F's requested works [1,2,3,7,8] mainly target **few‑step accelerations** via discretization‑schedule optimization, which is only **tangentially relevant** to our main focus. These techniques typically **match or worsen** the best‑FID of their baselines at large NFE.
>
> In contrast, our method’s main contribution is to improve the **score‑matching trajectory itself via the learned matrix schedule $M_t(\theta)$**, which allows us to improve **the best‑FID at high NFE**.
>
> Finally, the prior fast samplers could be applied on top of our learned $M_t(\theta)$ trajectory, making our approaches **complementary rather than competing**.
>
>
> ### Evaluation of larger models/data
>
> > **wK9F:** Asks how well the proposed method scales to larger models and datasets, and requests at least preliminary results on ImageNet 64×64 or ImageNet 256x256.
>
> **◆ In additional Imagenet 64x64 experiments, our methods again beat [1,2,3] in best-reported-FID by a clear margin (data [here](https://openreview.net/forum?id=YlbrXb2c1q&noteId=HxFfNGwrQm)) and our model beats the  baseline at low NFE.**
>
> We note that reviewer wK9F made this request **4 days before the end of discussion period**, so we did not have time for more comprehensive evaluations / ImageNet 256x256.
>
>
> ### Some ablations setups degrade FID
>
> > **wK9F:** $g^{ani}_1, g^{ani}_2, g^{iso}$ degrade FID (Table 1 of paper); Only learned discretization seems to provide consistent gains in FID.
>
> This concern arises from a misunderstanding of the roles of the anisotropic schedule $M_t(\theta)$ and the learned discretization $r(t;\gamma)$.
>
> ◆ Our full method learns a **separate scalar time transform** $r(t;\gamma)$ specifically to minimize discretization error, and then **composes** it with the learned $M_t(\theta)$.
>
> See $g\^{iso}_w$, ($g\^{ani}_{1,w},g\^{ani}_{2,w}$) in Table 1. These significantly improve FID across all three datasets (CIFAR‑10, AFHQv2, FFHQ; see Table 1 and Fig. 5). Note that the discretization schedule is **never used in isolation**.
>
> ◆ $g\^{ani}_1, g\^{ani}_2, g\^{iso}$ are **ablation studies**, expected to do badly
> In these runs, the matrix schedule $M_t(\theta)$ is (mis)used for discretization as well (**outside its intended role**), even though $M_t(\theta)$ is optimized for only **score-matching**. They serve to highlight the importance of composing $M_t(\theta)$ with $r(t;\gamma)$.
>
> ### Request for improved FID Evauation
>
> > **wK9F** asks for minimum of FID over three sets of seeded 50k images
>
> We updated Table 1 of our paper to report this number. We report the additional comparison to [3] over three runs [here](https://openreview.net/forum?id=YlbrXb2c1q&noteId=HxFfNGwrQm). In short:
>
> - FIDs of all methods improved slightly, but relative *ordering* **did not change**.
> - The relative *gap* changed slightly **in favor of our method**.

---

> ### Author Response · Authors · 2025-12-03
> **Rebuttal Summary (5/5 Response to FM4s)**
>
> Reviewer **FM4s** has two main concerns: (1) DCT basis is a restrictive form of anisotropy, (2) Unclear what $p_t$ should be at optimality.
>
>
> ### Restrictions on anisotropy / basis choice
>
> > **FM4s:** The anisotropy is restricted to a fixed DCT basis...It would be better if this basis can be learned instead. It would be interesting to see how this method performs on other bases ... for example the wavelet basis.
>
> Our framework for optimizing $M_t(\theta)$ applies to **arbitrary parameterizations of $M_t(\theta)$**, and *also* supports **learning of the basis itself**.
>
> We chose the DCT basis for simplicity. We also provided additional experiments comparing DCT, PCA, and Haar wavelet basis (see [here](https://openreview.net/forum?id=YlbrXb2c1q&noteId=cUzQbJeZ9S)). Our result demonstrates that
>  - Our method **generalizes well to different basis choice**
>  - Different basis (e.g. PCA vs DCT) are better at different NFE regimes, highlighting the potential for using our method to **learn a combination of these bases**.
>
> ### Interpretation of $p_t$ at optimality
>
>    > **FM4s:** It is unclear what $p_t(x; \theta)$ should look like at optimality.
>
>    We clarify that $p_t = p_0 * \mathcal{N}(0,M_t(\theta))$ where $M_t(\theta)$ minimizes the trajectory-level score matching error. Figures 2, 3 of our paper shows how $M_t(\theta)$ denoises more aggressively along certain subspaces.
>
>    We cannot provide a closed-form expression for the optimal $M_t(\theta)$, because the TLSM objective depends on `net` in a complex way. However, on an intuitive level, denoising certain subspaces quickly improves score-matching for other subspaces later on.

---

> ### Author Response · Authors · 2025-12-03
> **references**
>
> [1] Align Your Steps: Optimizing Sampling Schedules in Diffusion Models (Sabour et al., 2024)
>
> [2] Fast ODE-based Sampling for Diffusion Models in Around 5 Steps (Zhou et al., 2024)
>
> [3] Accelerating Diffusion Sampling with Optimized Time Steps (Xue et al., 2024)
>
> [4] Improved Denoising Diffusion Probabilistic Models (Nichol & Dhariwal, 2021)
>
> [5] Simple ReFlow: Improved Techniques for Fast Flow Models (Kim et al., 2025)
>
> [6] Consistency Models Made Easy (Geng et al., 2024)
>
> [7] DPM-Solver: A Fast ODE Solver for Diffusion Probabilistic Model Sampling in Around 10 Steps (Lu et al., 2022)
>
> [8] Fast Sampling of Diffusion Models with Exponential Integrator (Zhang & Chen, 2023)

---

### Meta-Review · Area_Chair_bXzG · 2026-01-05

**Summary:**

1. Is the proposal to have matrix-valued schedules novel? (mMms, wK9F)
2. A comparison to methods dedicated to reducing NFE was missing (wK9F).
3. Decoupling optimization of M_t and r(t, gamma) is heuristic and greedy, so the method might not be trajectory-optimal in the end (mMms).
4. Experimental gains were marginal (wK9F).

**Reviewer Concerns:**

1. This was clarified.
2. The authors added that overall, competing methods outperform theirs at low NFE. Gains are to be expected for higher NFEs.
3. This was partly clarified; the training does not assume any fixed discretization, but targets an improvement regardless of NFE.
4. Experimental gains are indeed marginal. Comparing with the original EDM paper (which takes more NFEs into account), an improvement is only achieved on FFHQ, from 2.39 to 2.31, which amounts to -0.08. On CIFAR10, EDM report 1.79 vs 1.93 here (+0.14), and on AFHQv2, EDM have 1.96 vs 2.02 (+0.06). While there may be a regime of intermediate NFE where the proposed method is better, this is not reflected in the framing of the paper. I encourage the authors to sharpen the experimental advantages in an updated version of this manuscript that includes low-NFE baselines, to which reviewer FM4s has already contributed some ideas. I also encourage the authors to absorb some of the technical details in the appendix to make room for an interpretation of the learned schedule.

**Reviewer Scores:**

- mMms's concerns were largely resolved as non-issues. It is hard to judge how the score would have changed in light of the clearer exposition of the experimental results due to the other reviewer's comments.
- wK9F was proactive in the discussion and already suggested that they still had reservations about the paper, in particular the experimental advantages of the proposed method. While the authors provided additional results, I think the reviewer would still have wanted to see stronger improvement to pass the bar of acceptance (see my comment on concern 4 above)
- I think FM4s would not have changed their scores. While they would have appreciated the additional results on changing the basis functions and the clarifications, I would think they would be sceptical about the best FID over all NFEs (see my comment on concern 4 above)

---

### Decision · Program_Chairs · 2026-01-26

Reject